# To Be Frail or Not to Be Frail: This Is the Question—A Critical Narrative Review of Frailty

**DOI:** 10.3390/jcm13030721

**Published:** 2024-01-26

**Authors:** Salvatore Sciacchitano, Valeria Carola, Giampaolo Nicolais, Simona Sciacchitano, Christian Napoli, Rita Mancini, Monica Rocco, Flaminia Coluzzi

**Affiliations:** 1Department of Clinical and Molecular Medicine, Sapienza University of Rome, 00189 Rome, Italy; rita.mancini@uniroma1.it; 2Unit of Anaesthesia, Intensive Care and Pain Medicine, Sant’Andrea University Hospital, 00189 Rome, Italy; monica.rocco@uniroma1.it (M.R.); flaminia.coluzzi@uniroma1.it (F.C.); 3Department of Life Sciences, Health and Health Professions, Link Campus University, 00165 Rome, Italy; 4Department of Dynamic and Clinical Psychology and Health Studies, Sapienza University of Rome, 00189 Rome, Italy; valeria.carola@uniroma1.it (V.C.); giampaolo.nicolais@uniroma1.it (G.N.); 5Department of Psychiatry, La Princesa University Hospital, 28006 Madrid, Spain; simona.sciacchitano@salud.madrid.org; 6Department of Surgical and Medical Science and Translational Medicine, Sapienza University of Rome, 00189 Rome, Italy; christian.napoli@uniroma1.it; 7Department Medical and Surgical Sciences and Biotechnologies, Sapienza University of Rome, Polo Pontino, 04100 Latina, Italy

**Keywords:** frailty, aging, epidemics/pandemics, risk factors, frailty pathogenesis, mitochondrial dysfunction, frailty assessment, interventions

## Abstract

Many factors have contributed to rendering frailty an emerging, relevant, and very popular concept. First, many pandemics that have affected humanity in history, including COVID-19, most recently, have had more severe effects on frail people compared to non-frail ones. Second, the increase in human life expectancy observed in many developed countries, including Italy has led to a rise in the percentage of the older population that is more likely to be frail, which is why frailty is much a more common concern among geriatricians compared to other the various health-care professionals. Third, the stratification of people according to the occurrence and the degree of frailty allows healthcare decision makers to adequately plan for the allocation of available human professional and economic resources. Since frailty is considered to be fully preventable, there are relevant consequences in terms of potential benefits both in terms of the clinical outcome and healthcare costs. Frailty is becoming a popular, pervasive, and almost omnipresent concept in many different contexts, including clinical medicine, physical health, lifestyle behavior, mental health, health policy, and socio-economic planning sciences. The emergence of the new “science of frailty” has been recently acknowledged. However, there is still debate on the exact definition of frailty, the pathogenic mechanisms involved, the most appropriate method to assess frailty, and consequently, who should be considered frail. This narrative review aims to analyze frailty from many different aspects and points of view, with a special focus on the proposed pathogenic mechanisms, the various factors that have been considered in the assessment of frailty, and the emerging role of biomarkers in the early recognition of frailty, particularly on the role of mitochondria. According to the extensive literature on this topic, it is clear that frailty is a very complex syndrome, involving many different domains and affecting multiple physiological systems. Therefore, its management should be directed towards a comprehensive and multifaceted holistic approach and a personalized intervention strategy to slow down its progression or even to completely reverse the course of this condition.

## 1. Introduction

### 1.1. What Does Frailty Mean and Who Are the Frail Elderly?

The term frailty is derived from the Latin term “fragilitas”, which refers to something weak that can be easily broken, ruined, or destroyed. Frailty was initially described in the insurance industry as a measure of the heterogeneity of the mortality risk of people over 65 years old [1]. The term was then adopted by clinicians to describe either dependent, institutionalized patients over 65 years old [2], or older patients hospitalized with multimorbidity and an increased mortality risk [3]. This concept has since evolved, and frailty is now described in many different domains, including physical, cognitive, social, emotional, as well as economic. However, the question raised by Woodhouse in 1988: “Who are the frail elderly?” remains without a definite answer [2]. Although frailty is a well-recognized cornerstone of geriatric medicine and has been associated with adverse health outcomes, its definition and the pathogenic mechanisms involved are still subject to intense debate. 

### 1.2. Frailty as a Syndrome with Impairments of Multiple Systems

Many different criteria have been proposed to describe the variable phenotypes observed in patients who show impairments in multiple systems. Physical impairment still represents the main aspect of frailty, but other conditions are relevant as well, including aging, genetic predisposition, gender, neurocognitive status, psychological well-being, socio-demographic variables, behavioral nutrition, and physical activity. Considering all the possible factors that are related to frailty, it lacks a clear definition. Frailty, in fact, is considered “one of those complex terms … with multiple and slippery meanings” [4]. Difficulties are especially related to the complex pathophysiology, the heterogeneous nature of the predisposing conditions that underline the frailty syndrome, the best method to measure frailty, and the most suitable criteria for its diagnosis. Due to these factors, a comprehensive diagnostic tool or test for frailty with biological elucidations is still lacking [5]. Considering the highly heterogeneous nature of this clinical syndrome, a multidisciplinary approach, including not only the evaluation of specific medical conditions, but also neurocognitive deficits and their potential impact on the quality of life, is required for an accurate assessment of a frail subject [6]. To further increase the complexity of the multiple dimensions of frailty, the criteria used may be different if they are applied to people living in a community, an outpatient setting, or hospitals. Therefore, there is a need for a comprehensive management of frailty, possibly at an individual level, as indicated by the European Commission [7]. Frailty has been historically linked to the various pandemic waves of communicable diseases that have afflicted humankind throughout the centuries. 

### 1.3. Frailty in the Era of COVID-19 Pandemic

Recently, frailty has been gaining increasing attention because of the COVID-19 outbreak that highlighted the clinical relevance of this condition to the outcome of this disease. There was a pre-existing interest in the study of frailty before the COVID-19 pandemic [8], but there has been an increase in the number of studies (25,023 articles) referring to the term “frailty” that have been published in the literature until December 2022, with a consistent increase during the last few years (Figure 1). In particular, the recent increase in the interest in frailty is noticeable due to the high number of citations (16,858) during the 2011–2020 period. The interest in this field remained constant in the years 2021 and 2022, where frailty was mentioned in 4535 and 4843 scientific articles, respectively (Figure 1). In addition, several international working meetings have been performed, focusing their attention on frailty, and many recommendations for best practice guidelines have been published by several scientific societies. The adequate assessment of frailty has enormous potential implications in public health since the presence of frailty has been associated with increased risk of adverse outcomes, including falls, hospitalization, admission to long-term care, and mortality, with a consequent exponential rise in healthcare costs [9]. More importantly, the acquisition of frailty is not an irreversible process and it can be prevented, delayed, or even completely reversed with appropriate pharmaceutical as well as non-pharmaceutical interventions [10]. The recent COVID-19 pandemic, with its high impact on mortality, provided the opportunity to study a large population in a unique natural experiment, and many studies have been conducted to investigate the correlation between frailly and the occurrence or the severity of COVID-19 [11]. At the beginning of the COVID-19 pandemic, we started a network project, involving six Italian institutions, to measure and compare the impacts of the pandemic on access to a cure and to establish a COVID-19 Biobank of biological samples to search for immune determinants associated with frailty in these patients [12]. Following this project, we initiated a clinical trial, founded by the Italian Minister of Health in the context of the National Recovery and Resilience Plan (NRRP) (Piano Nazionale di Ripresa e Resilienza, PNRR, mission 6 (health), as part of the Next Generation EU (NGEU) program, focused on the analysis of the pathogenic basis of frailty, with special attention to the evaluation of energetic metabolism in hospitalized patients. One of the most difficult challenges in planning these projects was to reach an agreement on a common definition of frailty. We realized that defining frailty is a complex and challenging task because of the uncertainty of its characterization. In order to classify a person as frail, we need to consider the subject from different points of view and in a dynamic perspective. 

### 1.4. Aim of This Narrative Review 

This narrative review aims to analyze the various factors that have been considered in the assessment of frailty, with a special focus on the proposed pathogenic mechanisms and the emerging role of biomarkers in the early recognition of frailty for timely intervention in order to prevent poor outcomes. This paper reviews the current state of knowledge regarding frailty by focusing on specific areas: (i) definitions of frailty, (ii) concept of frailty in the pandemics, (iii) prevalence and incidence of frailty, (iv) risk factors for frailty, (v) various pathogenic mechanisms of frailty, (vi) biomarkers of frailty, (vii) frailty and diseases, (viii) methods to assess frailty, (ix) European guidelines, (x) intervention to reduce frailty, (xi) frailty and implications for policy and practice, and (xii) frailty and digital health.

## 2. Definition of Frailty in History

Frailty was initially considered synonymous with the concepts of institutionalization [13] and with the concepts of disability, comorbidity, and failure to thrive in elderly people [14]. The frailty designation was thus used interchangeably to identify a group of older adults with physical vulnerabilities [15]. The recognition that frailty should be considered distinct from disability and comorbidity and the introduction of some well-established frailty assessment tools, has led to a better definition of frailty: “biologic syndrome of decreased reserve and resistance to stressors, resulting from cumulative declines across multiple physiologic systems, and causing vulnerability to adverse outcomes” [9]. 

### 2.1. The Science of Frailty

Based on such assumptions, frailty was considered a new clinical entity that deserved much attention, and numerous studies have been conducted to increase the overall clinical impact of the newly acknowledged “science of frailty”. However, although there have been several attempts to describe frailty, there is still a lack of a unique and generally accepted definition [16], and its definition remains controversial [17]. As previously noted, the syndrome should be kept distinct from disability, functional dependency, sarcopenia, and/or multimorbidity. Under these conditions, the subject is unable to perform a task without help from someone else or the subject may have simply been afflicted by several illnesses [15]. A frail subject can certainly be disabled, may have sarcopenia, and/or may be affected by multiple diseases, but these conditions represent only specific aspects of the medical syndrome of frailty and do not thoroughly capture the whole concept of frailty [18]. The condition of frailty should imply that the subject is able to perform an activity but needs assistance to perform it. A certain agreement regarding the definition of frailty was reached in a survey regarding its position as a stage halfway between robust and disability [19]. It is difficult to develop a precise definition because frailty represents a medical syndrome [20] that involves multifaceted aspects [21]. 

### 2.2. Multiple Areas and Domains of Frailty

There are multiple areas and domains where the term “frailty” has been used to define specific conditions of vulnerability (Table 1) (Figure 2). In support of this view, manifestations of frailty consist of the aggregation of symptoms in a critical mass and in a hierarchical order, where dysregulation in a sentinel system is able to trigger a cascade of alterations across other systems. It is thus possible to justify the current strategy that subdivides frailty into categories ranging from the absence of frailty and progressively increasing in severity to the highest level of frailty (i.e., non-frail, pre-frail, and frail). However, despite many international operative consensuses [22], many WHO reports [17], and recommendations [23], an agreement has not yet been reached, and we are still in search of a gold standard definition of frailty [24,25]. Due to the diversity of the parameters affecting the development of frailty, the term “frailty puzzle” has been proposed [26]. Due to the difficulties in trying to find the best definition of frailty, there is currently only a negative definition for the description of a frail person. According to the Collins English Dictionary, a frail person can be defined as someone who “is not very strong or healthy” [27].

## 3. The Concept of Frailty in the Pandemics

The occurrence of pandemics of communicable diseases has often played a role in human history, often having massive lethal impacts [28]. The recurring outbreaks of many different plagues have often represented a dramatic and crucial turning point in the societies that were affected [29]. In some cases, their occurrence has been related to climate changes [30]. However, they also seem to be connected to the growth of civilizations and expansion of empires in different parts of the world, which has often been accompanied by more opportunities for the spread of infectious diseases. This phenomenon did not only occur in ancient times, but it has also been observed in recent periods, with an increase in dangerous outbreaks around the world [31]. The number of new diseases per decade has increased nearly fourfold over the past 60 years, and since 1980, the number of outbreaks per year has more than tripled. The occurrence of pandemics in waves has been well represented in the painting “The Great Wave off Kanagawa” by the Japanese painter Katsushika Hokusai, where the frailest individuals were depicted as fishermen in a boat, struggling against overwhelming forces of the sea’s waves [32,33] (Figure 3). There have been numerous attempts directed to answer the question of whether plagues had to be considered as indiscriminate, random, mass killers, or whether they should be viewed as focused killers. In other words, did they have specific and greater effects on the frailest individuals among the affected population [34]? The question may also be considered in this way: Should frailty be considered as an intrinsic characteristic of individuals, in a sort of innate risk, or should it be considered as the combined effect of various external conditions? For the ancient plagues, the only possible approach to answer this question was to recover and collect bio-paleopathological samples of people who died and were buried during the historic pandemic waves and to assess frailty in their bones by comparing the results with those obtained in skeletons from coeval subjects who died as a result of other factors. The assumption that frailty measured in bones, the so-called “skeletal frailty”, can be used as a proxy for the overall health status of past populations is based on several previous observations reported by bioarchaeologists [35,36,37,38,39,40].

### 3.1. The Black Death

The Black Death pandemic is one of the most devastating moments in human history. It was caused by the spreading of the infection by the bacterium *Yersinia pestis*. It is estimated that up to 200 million people died because of the plague pandemic. The bubonic plague came in three subsequent waves. The first was in the Mediterranean region in 541 A.D. and is known as the Plague of Justinian. The second wave hit Europe between 1348 and 1350 and is known as the “Black Death”. In the 1800s, the plague reappeared as a third wave in China. The approach based on the measurement of skeletal frailty in subjects was applied to the analysis of subjects deceased because of the Black Death pandemic more than 700 years ago. The bones retrieved from one cemetery, located near London, indicated that Black Death mortality was rather selective since it affected more people with skeletal frailty [41,42], older subjects [43], and women [44]. However, such results were not confirmed by the analysis of bones retrieved from one “lazzaretto” (the Italian name for a hospital for plague victims), located in Imola, Italy [34]. Since healthier individuals (i.e., with low levels of skeletal frailty) could be found only among plague victims, the authors concluded that the plague did not affect only frail individuals [34]. Recently, the analysis of the DNA obtained from the teeth of 206 ancient skeletons of individuals who died due to the Black Death revealed mutations in a specific gene, namely the ERAP2 gene. Such mutations could have been associated with greater chances of surviving the plague [45]. The mutation of this gene was responsible for variations in cytokine response to *Yersinia pestis* and increased the ability to control intracellular *Yersinis pestis* by macrophages. This mutation was later transferred to the following generations and is now part of our DNA. However, this protective mutation overlaps with alleles that are associated with increased susceptibility to autoimmune diseases, such as the inflammatory bowel disease, Crohn’s. In other words, the mutation that helped our ancestors survive 700 years ago could be responsible for the damage to our health today.

### 3.2. The Russian Flu Pandemic of 1889

The Russian flu pandemic of 1889 is the first pandemic to be statistically analyzed and the first epidemic to be widely addressed by the developing daily press, receiving extensive media coverage [46]. The causative agent was identified as *Myxovirus influenzae*, but this assumption was recently questioned, and it was hypothesized that this pandemic could have been caused by a Coronavirus [47]. The rapid spread of this flu was surprising, and it can be attributed to its high level of contagiousness and its transmission by human carriers traveling by railroads and steamboats, which were under constant expansion across Asia and Europe during that period. The mortality rate was highest among young adults (under 50 years of age), whereas influenza usually affected the oldest and the youngest. However, it has been reported that the Russian flu was more common in men, and its clinical course was particularly severe in the elderly [48,49,50] or patients with comorbidities, including phthisis, cardiopathies, cerebral diseases, and nephritis.

### 3.3. Pandemics of the 20th Century

Many different waves of pandemics affected people worldwide in the 20th century. The dissemination of viral infections was also facilitated by globalization, which accelerated human integration and interconnection, thus promoting infectious diseases [51]. There have been at least three major influenza pandemics in the past hundred years [52].

#### 3.3.1. The Spanish Flu

The one that was marked by high virulence and fatality was the Spanish flu pandemic of 1918 [53,54]. The causative agent was identified as an H1N1 virus, present in wild waterfowl birds, the main reservoir of such virus, with pigs serving as intermediate hosts in the transmission from birds to humans [55]. The spread of the virus across the globe was facilitated by World War I military troops from different countries who served as carriers. The pandemic is known as the “Spanish Flu Pandemic” because Spain was the first to acknowledge the disease outbreak [56]. The pandemic was responsible for millions of deaths across the globe, although the exact number is controversial [53,54,57,58]. The singular aspect of this pandemic is that mortality was high among young subjects. The general behavior of other pandemics is that the frailest infected subjects, namely the young (<20 years of age) and the adults (>40 years of age), are those that are affected more and that show higher mortality, with a typical “U-shaped” curve of mortality. On the contrary, the Spanish flu, caused by the H1N1 virus, was responsible for the occurrence of an additional mortality peak among people between 20 and 40 years old, thus showing a “W-shaped” curve [59,60]. This was described as the frailty–mortality paradox of the Spanish flu [61]. To investigate the impact of this pandemic, the same bio-paleopathological approach used in the analysis of the Black Death was applied to analyze skeletal lesions in individuals who died during the Spanish Flu pandemic of 1918. Mortality data, obtained from the Hamann-Todd Osteological Collection at the Cleveland Museum of Natural History, indicated that this pandemic, characterized by a high case fatality rate of 2.5%, affected mostly healthy young adults, generally considered the least frail segment of the population. The vulnerability of young adults associated with the 1918 pandemic was observed in France, Italy, Spain, and Portugal [62]. The exact reason for this reported paradox is not known and is considered one of the great mysteries of the 1918 pandemic [63]. One possible explanation was that young people died more because of an over-reaction of their active immune systems, referred to as a cytokine storm, but this theory has not been proven yet. The 1918 Spanish flu was the first of three flu pandemics caused by the H1N1 influenza A virus [64]. The most recent one was the 2009 swine flu pandemic [65]. Another pandemic caused by the H1N1 virus was the 1977 Russian flu [66].

#### 3.3.2. The Asian Flu

Another strain of the Influenza A virus, namely the H2N2 virus, emerged in February 1957 in East Asia, and the infection traveled to other regions of the globe by the summer of the same year. The outbreak was mild compared to the previous Spanish flu and was named the Asian flu pandemic [67]. Similar to H1N1 virus infections, the outbreak appeared in different phases of the waves but was less transmissible. It caused approximately 1.1 M deaths worldwide from the year 1957 to 1958 [68]. It caused many infections in children, and it was most deadly in pregnant women, in the elderly, and in those with pre-existing heart and lung diseases [68].

#### 3.3.3. The Hong Kong Flu

A decade after the outbreak of the H2N2 virus, due to genetic reassortment, the H3N2 virus evolved in 1968, leading to the Hong Kong flu [69]. This flu pandemic lasted until 1969–1970, and although it was associated with comparatively few deaths worldwide, the high level of contagiousness of the virus facilitated its rapid global dissemination. 

### 3.4. HIV/AIDS

HIV remains a major global public health issue, having claimed more than 40 million lives so far, with an estimated 39 million people living with HIV at the end of 2022, two thirds of whom are in the WHO African Region [70]. Frailty negatively affects the People Living With HIV (PLWH)’s clinical status and increases their risk of adverse outcomes, impacting their quality of life and health span [71]. It has been reported that adults living with an HIV infection experience a prevalence of frailty equivalent to, and even greater than, that observed in elderly people [72,73,74]. It is not clear whether the occurrence of frailty is responsible for a predisposition to HIV infections or if it is a consequence of the HIV infection itself, suboptimal medication, or control of the infection. In this regard, it should be underlined that HIV infections, with a typical drop in CD4 cell count, render the immune system unable to protect against both infectious and chronic diseases such as cancer, cardiac diseases, and diabetes, thus contributing to the occurrence of physical deterioration and frailty. In this regard, HIV and COVID-19 showed a similar, but simultaneously distinct, occurrence of lymphocytopenia [75]. However, the availability of antiretroviral medications has played a major role in the dramatic demographic shift that occurred between 2000 and 2015, where AIDS-related deaths fell by 28%, with a relevant increase in HIV infections in people aged 50 years or more [76]. It has been reported that early control of HIV replication by combination antiretroviral therapy (cART) and the consequent virological suppression and CD4+T-cell recovery is able to decrease the risk of developing frailty [77]. Adequate treatment of HIV infection, therefore, has the potential to prevent early aging, excess morbidities, and early death associated with HIV infections. For these reasons, major attempts are currently directed to screen, assess, and ideally reverse frailty in people with HIV [78]. 

### 3.5. Pandemics of the 21st Century, the Century of the Coronavirus 

The impact of the coronavirus infection on global public health has dramatically changed our views about this family of viruses. They were considered to be pathogens, most typically associated with the common cold, and thought to be responsible for mild to moderate upper-respiratory tract illnesses in humans. However, the emergence and reemergence of viral outbreaks of SARS-CoV in 2002, MERS-CoV in 2012, and more recently, SARS-CoV-2 in 2019, have led to very high fatality rates, especially among frail people and have directly impacted healthcare systems worldwide, undermining global sustainable development goals [79,80].

#### 3.5.1. Severe Acute Respiratory Syndrome (SARS)

The severe acute respiratory syndrome (SARS) was caused by the coronavirus named SARS-CoV and originated in Guangdong province (China) in 2003. The possible natural reservoir of this virus was identified in bats [81], and palm civets could be the intermediary hosts before dissemination to humans [82]. One of the characteristics of the SARS epidemic was its striking age-dependent case-fatality ratio that affected most old people > 65 years old (>50%) compared to young people < 24 years (<1%), suggesting that age was a major factor in frailty [83,84,85].

#### 3.5.2. Middle East Respiratory Syndrome (MERS)

After ten years, a new coronavirus, in Jeddah, Saudi Arabia, was responsible for the emergence of another epidemic, MERS-CoV. Once more, bats were identified as the potential animal reservoirs for this virus, and dromedary camels were considered to be the intermediary hosts [86]. Frail people affected by comorbidities such as diabetes, kidney disease, chronic lung disease, and cancer, or individuals on immunosuppressive treatments were considered at high risk of developing severe symptoms due to MERS-CoV, and were advised to avoid close contact with camels and bats. The epidemic started in the Middle East but affected many countries, and according to the WHO, deaths were reported in as many as 27 different countries [87].

#### 3.5.3. Coronavirus Disease 2019 (COVID-19)

Coronavirus disease 2019 (COVID-19) is caused by the severe acute respiratory syndrome coronavirus [88,89] and then quickly spread worldwide, resulting in the declaration of pandemic by WHO [90]. It became an international health emergency, which was responsible for one of the most life-threatening global pandemics in history, with a total of 767,518,723 confirmed cases during the various waves and 6,947,192 deaths, according to the WHO [91]. The COVID-19 pandemic triggered a significant increase in mortality and a reduction in life expectancy in several countries [11,92]. The clinical presentation of the disease was variable, with some patients reporting only mild-to-moderate symptoms such as fever, persistent dry cough, body aches, and occasional breathlessness. In other patients, the disease was responsible for acute respiratory failure and acute respiratory distress syndrome, with associated sepsis or multiorgan failure [93]. Similar to historical pandemics, the difference in the severity of the disease in the affected patients was related to the occurrence of frailty. Many studies reported that COVID-19 patients with frailty had an increased risk of short-term mortality compared to non-frail patients with COVID-19 [94,95,96,97,98].

## 4. Prevalence and Incidence of Frailty

The incidence and prevalence of frailty in a given population are difficult to assess. It seems that frailty is common among community-dwelling older adults [99,100], as well as in nursing homes [101]. Approximately 8.5% of the global population is 65 years old or older, and this percentage is projected to increase to nearly 17% by 2050 [102]. It is generally assumed that the prevalence of frailty increases with age, is higher in women than men, and is more prevalent in the presence of chronic diseases (Table 2). However, there is no consensus about the exact prevalence rates of frailty in the general population. A global estimate of frailty ranged from 3.5 to 27.3% [103]. The incidence of frailty has been reported to be 71.8/1000 person years in the Cardiovascular Health Study (CHS) [9]. In the Precipitating Events Project, it was reported to be 22.5 to 38.7/1000 person years [104]. In the Women’s Health and Aging Study-I, it was reported to be as high as 191/1000 person years in disabled women living in the community [105]. A systematic review and meta-analysis of literature on frailty prevalence in 22 European countries was conducted in 2018 [106]. In this systematic review, the overall prevalence of frailty was 18% (Table 3). We should consider that the data obtained by the reported review and meta-analyses of literature reflect the heterogeneity in the methodological approaches used. In the metanalysis by Kojima et al., the mean prevalence of frailty, measured in long-term (nursing home) care (LTC), ranged from 19% to 75.6% [101]. In countries in Latin America and the Caribbean (LAC), which have a high prevalence of chronic and incapacitating diseases, one in five older adults was considered frail [107]. The prevalence of frailty also depends on the context in which it is measured. In the USA, approximately 15% of noninstitutionalized adults are frail [108]. In adult patients seen in ambulatory settings, it ranged from 4% to 59% [99]. In patients admitted to intensive care units (ICUs), the prevalence of frailty was 38.6% [109]. Frailty prevalence is also influenced by gender, race/ethnicity, and socioeconomic status. It has been reported that frailty is more common in women than in men (17.2% vs. 12.9%), in African Americans compared to non-Hispanic whites (22.9% vs. 13.8%), in Hispanic Americans compared to non-Hispanic whites (24.6% vs. 13.8%), and in lower income compared to higher income groups (25.8% vs. 5.9%) (108). Once again, the diagnostic parameters used for the definition of frailty varied substantially in the different studies considered, with the highest frequencies observed in studies that used multidimensional instruments to evaluate this construct [99].

## 5. Risk Factors for Frailty

The identification of risk factors for frailty represents a challenging task for clinicians, epidemiologists, and scientists. Many different risk factors or conditions involving various cellular and molecular aging processes, alone or in combination with environmental, genetic, and chronic disease states, have been reported to increase the risk of frailty, including alcohol excess, cognitive impairment, falls, functional impairment, hearing problems, mood problems, nutritional compromise, physical inactivity, polypharmacy, smoking, vision problems, social isolation, and loneliness. They are thought to act together to drive the development of frailty [24,110,111]. Since many different criteria used to assess physical frailty phenotype reflect muscle health, it is reasonable to conclude that sarcopenia (muscle loss, weakness, and reduced muscle function, frequently associated with aging) represents one of the major risk factors for frailty [112,113]. However, a more comprehensive approach was introduced by the WHO in 2015, in which the risk of frailty was associated with the combination of both physical and psychosocial capacities [114]. Early identification of risk factors has relevant consequences in preventing the various complications related to the occurrence of frailty and offers the opportunity to encourage and support guidance on promoting healthy aging. In this regard, various organizations have furnished guidelines to encourage healthy aging and to postpone or even avoid the onset of frailty [115]. A partial list of resources available from various organizations is reported here:Skills for Health, NHS England, and the Health Education England Frailty Framework of Core Capabilities [116]NHS England Practical guide to healthy aging [117]Age UK advice on keeping active and aging well [118]NHS Choices provides advice on how physical activity and exercise can help people stay healthy, energetic, and independent as they grow older [119]NICE supporting guidance on healthy aging [120]Public Health England guidance on productive healthy aging and musculoskeletal health [121]The Academy of Medical Royal Colleges has published “Exercise: The miracle cure and the role of the doctor in promoting it” [122]CDC’s National Center for Chronic Disease Prevention and Health Promotion [123]Italian National Center for Disease Prevention and Health Promotion [124]Gaining Health: Making Healthy Choices Easier Italy [125]

## 6. Various Pathogenic Mechanisms of Frailty

There is no consensus regarding the mechanisms responsible for the acquisition of a frail phenotype, and different hypotheses and conceptual models of frailty have been proposed in the past. Three components were initially considered central to frailty: neurologic control, mechanical performance, and energy metabolism [126]. Subsequently, frailty was described as a syndrome and was separated by disability [127]. According to this outlook on frailty, five components were interlinked to form a “cycle of frailty”, namely weakness, slowness, exhaustion, low activity, and weight loss [128]. However, it soon became clear that the situation is more complex and there is still work to be conducted regarding the hypotheses and theories until more precise criteria and definition of the pathogenic mechanisms of frailty can be developed. According to the theories on the physiological reserves of the human body, frailty may develop as a consequence of the loss of adaptability to physiologic stressors, known as the “vulnerability to stressor” theory [126]. Frailty may also occur when there is a discrepancy between energy metabolism and requirements, which means that in order to maintain a strong ability, one must practice it, otherwise they risk losing it, known as the “use it or lose it” theory [129]. Similar to the latter, there is also a theory based on the progressive decline in physiological reserves from underlying aging biological processes leading to a decreased capacity to respond to stress, and a clinical vicious cycle of energy dysregulation, known as the “cycle of frailty” theory [9]. In the same group of theories based on physiological reserves of the human body, it is important to cite those concerning the loss of adaptability, leading to dysfunctions in many different organs and systems, known as the “loss of complexity” [130], “dynamical system” [131], and “homeostatic dysregulation” theories [132]. According to other theories, frailty should be viewed as the result of progressive and age-related cumulative deficits that can be measured as symptoms, signs, functional impairments, and laboratory abnormalities [133]. Finally, another hypothesis to explain the occurrence of frailty is related to pre-disability, which considers frailty as a pre-disability condition [134]. The early recognition of frailty may thus help to prevent progression toward the acquisition of a permanent disability. Based on the observation that oxidative damage is correlated with frailty in both humans and experimental mouse models, a “free radical” theory has been postulated [111,135]. The demonstration of the relevant role of mitochondria in the production of free radicals has recently led to the reconsideration of this hypothesis, which was renamed to “mitochondrial free radical theory of ageing (MFRTA)” [136].

### 6.1. Relevance of Genetic Predisposition to Frailty

Since the early detection of frailty may allow the prevention or even reversion of the progression toward the acquisition of a frail phenotype, the identification of genetic markers of predisposition to frailty would allow an earlier and more objective recognition of frail individuals. Therefore, many efforts have been made to identify gene mutations, different types of genetic damages, and damages in the cellular repair capacity that may facilitate the occurrence of frailty. However, several studies performed in older adults, classified into frail, prefrail, and non-frail failed to detect specific gene mutations associated with frailty [137]. In the same study, a nonsignificant association was found between decreased DNA repair capacity and frailty, and persistent levels of phosphorylated H2AX, as indicative of DNA breakage, increased progressively with frailty severity [137]. Another approach consisted of the genetic analysis of twins [138,139]. The higher intraclass correlation of frailty measures in monozygotic twins compared to dizygotic twins pointed towards underlying genetics of frailty [140]. In agreement with the hypothesis that a genetic predisposition to frailty does exist, an extensive database search was recently performed [141]. Biomarker panels for frailty would be of higher value and represent better alternatives to single markers for the complex genetic predisposition to frailty. Based on this extensive search, there was no clear association with single specific marker genes. Rather, a wide collection of genes, grouped in panels of frailty biomarkers, have been proposed. The core panel identified consisted of 19 high priority candidate genes in seven groups, namely (1) IL-6, CXCL10, and CX3CL1 (2) GDF15, FNDC5, and VIM, (3) regucalcin and calreticulin, (4) PLAU and AGT, (5) agrin, BDNF, and progranulin (6) alpha-klotho, FGF23, FGF21, and leptin, and (7) miRNA panel (to be further defined), AHCY, and KRT18. These authors also indicated that an expanded panel would include additional genes, potentially related to frailty, grouped in medium as well as low-priority clusters [141]. Studies performed on twin brothers in the UK and Denmark suggested a possible role of genetic components in increasing the risk of frailty [142,143], with a heritability estimate of 25% and 30%, respectively, for frailty defined using syndromic and cumulative deficit frailty index [142]. This result is comparable to that obtained for participants aged 60 and older, who were enrolled in the Framingham heart study, where the heritability of the combined trait of prefrailty and frailty vs no frailty was estimated to be 19% [144]. The genetic underpinning of frailty was analyzed by Genome Wide Association Studies (GWAS) for the first time in 2018 [145]. Since then, other studies have been performed, and in some of them, a possible link between frailty and longevity was suggested [146]. Two genes have shown a consistent association with the longevity phenotype, namely the APOE gene [147,148] and the FOXO3 gene [149,150]. However, the complexity of frailty has made it difficult to determine its underlying genetics and to replicate such genetic observations in frail subjects. A more recent GWAS study identified a total of 37 independent and novel loci that were associated with frailty, in particular with the Fried Frailty Score in a cohort of over 380,000 participants of European ancestry [151]. Most of them have been previously reported in GWAS of other traits, including obesity/BMI, lipids, coronary artery disease, hypertension, diabetes, and cancer, reinforcing the concept that these conditions are strictly connected with frailty. The genetic analysis by a genome-wide association study, involving 1980 patients with Covid-19 and severe disease (defined as respiratory failure) at seven hospitals in the Italian and Spanish epicenters of the SARS-CoV-2 pandemic in Europe, identified a 3p21.31 gene cluster as a genetic susceptibility locus in patients with COVID-19 with respiratory failure and confirmed a potential involvement of the ABO blood-group system [152]. Following this observation, an interesting correlation was observed between genetic loci that confer a genetic predisposition to frailty and the core haplotype that has been inherited from Neanderthals [153] in COVID-19 patients from the province of Bergamo, Italy, considered one of the deadliest zones affected by the virus in the Western world, and similar results have been reported by other researchers [154]. In the loci that have been inherited from Neanderthals, there are some genes, such as CCR9, CXCR6, and LZTFL1, already identified in the study published in the NEJM [152], that can be detected in a relevant percentage of European (16%) and many Asiatic subjects (50%).

### 6.2. Frailty and Socioeconomic and Demographic Factors

Most of the studies related to the occurrence of frailty in many different countries and many different ethnicities have been focused on factors associated with progressive compromise of biological health. There is little evidence of the influence of socioeconomic factors on the occurrence of frailty. While approximately 10% of older adults are living with frailty in economically developed countries, little is known about frailty prevalence and the nature of frailty in different ethnic groups living in low- to middle-income countries. Some studies have observed that frailty prevalence was higher in adults in upper-middle income countries and low- to middle-income countries, compared to those living in high-income countries [155,156]. The same difference was reported in countries where many ethnically diverse populations live in different socioeconomic conditions, as a result of migration forced by climate change and conflicts in an increasingly globalized world. In this regard, indigenous ethnic minorities, and marginalized groups living in economically developed countries such as the US, Australia, and New Zealand, have a higher prevalence of frailty than the white majority population [157]. Economically developed countries, primarily in the West have faced unprecedented levels of migration in the 20th and 21st centuries. Consequently, ethnic minorities face significant health inequalities that are becoming both persistent and pervasive [158]. Based on these observations, it has been suggested that the risk factors could be identified as the following: female sex, lower economic status, lower education levels, and belonging to migrant minority ethnic groups.

### 6.3. The Role of Aging

Approximately 8.5% of the global population is 65 years old or older, and this percentage is projected to increase to nearly 17% by the year 2050 [159,160]. Recently, in Japan, the national data showed that 29.1% of the 125 million population is aged 65 or older and that people aged 80 or older reached a record rate of more than one in 10 people [161]. In Italy, there is a clear demographic trend toward the aging of the general population. As of 1 January 2022, the aging index—the number of elders (at least 65 years old) per 100 persons younger (equal or less than 15 years)—was equal to 187.9%, an increase of more than 56 percentage points, compared to 20 years ago [162]. We expect a further increase of more than 100 percentage points in the next 20 years, with the aging index reaching 293%. Data from the 2022 report of the “Istituto Nazionale di Statistica” (ISTAT) showed that a total of 14 million and 46 thousand were registered as people of at least 65 years of age, three million more than that reported 20 years ago and five million less than the expected rate in 20 years [162]. The same trend is anticipated for those of more than 80 years of age, with more than 4,5 million people over 80 years old and 20,000 for the ultra-centennials at the beginning of 2022. Both of these populations are expected to increase. The aging process is associated with a progressive homeostatic and homeodynamic dysregulation responsible for the loss of resilience capacity, increasing the individual’s susceptibility to develop or have a worse frailty status [163]. Frailty is closely linked to aging, and increased chronological age is associated with frailty [99]. Many studies have highlighted the close correlation that exists between age and frailty [9,18,24,164]. Based on the systematic review of many different observations, it can be definitively asserted that the increased chronological age predicts frailty [99]. Elderly patients decline as they advance in age. The aging process is accompanied by the slowing down of movements, loss of strength, and reduction of mental acuity and abilities. All of these are part of the physical and psychological frailty syndromes. In addition, aging is paralleled by the accumulation of diseases and the cumulative decline and malfunction of many organs and systems. The frailty phenotype thus develops as a consequence of age-related decline in multiple systems. Everyone experiences frailty at a certain point in his/her life. The progressive accumulation of damage in various organs and systems results in a vulnerability to sudden health status changes that can be triggered even by relatively minor stressor events, the classical “straw that breaks the camel’s back”. In this regard, the “geroscience hypothesis” states that frailty and disability (the main characteristics of aging) are caused by a progressive discrepancy between the accumulation of damages in various organs and systems and host resilience [165]. This hypothesis has highlighted the importance of the link between aging and multiple diseases, and it has been stated that any intervention aimed to retard the aging process will simultaneously result in a delay in the onset of associated diseases. However, chronological age per se cannot be considered synonymous with frailty. Several studies have reported a highly heterogeneous distribution of the different frailty phenotypes (non-frail, pre-frail, and frail) within groups of the same age [166]. Non-frail patients have been found at approximately the same percentage (3.5% and 5.5% respectively) in two Portuguese studies comprising the oldest elderly subjects, namely those aged 80 or older [167] and the centenarians [168].

### 6.4. Frailty and Subjective Perception of Aging

Therefore, despite strong evidence that frailty correlates with aging, it must be acknowledged that age is not always related to frailty. Frail adults may feel older than their chronological age because of low physical activity and faster decline in their physical, psychological, and cognitive performance [169,170]. On the contrary, simple chronological age is not appropriate to classify a person older than 65 as frail. In countries where the life expectancy is 80 years, like in Japan and Italy, there is an increased number of bright and energetic elderly people, indicating that factors other than chronological age may play a major role in affecting health status and life expectancy. In such conditions, a more careful assessment of the “elderly frailty” is required to determine the vulnerability of these subjects [171]. It is thus relevant to distinguish between chronological and subjective age [172]. In this regard, subjective perception of aging does not necessarily coincide with the chronological age [173] and subjective age can predict an individual’s health condition better than simple chronological age [174,175,176,177,178]. 

### 6.5. The Importance of “Inflammaging”

The observation that neither old age nor disability alone identifies those at highest risk of adverse outcomes and that frailty is predictive for adverse effects independently of these factors [9,179] has led to the hypothesis that other factors may be pathogenetically relevant, including decreased musculoskeletal functioning, neurologic control, lower energy metabolism, as well as physiological alterations in immune and endocrine systems. There has been particular interest in the possible role of inflammation and the correlated clotting process, including the following inflammatory markers: C-reactive Protein (CRP), factor VIII, and fibrinogen, which are significantly associated with the syndrome of frailty, even in the absence of the most prevalent chronic diseases, including diabetes and cardiovascular diseases (CVD)s [180]. Moreover, considering the relationship between inflammation and hemostasis, the same significant association was reported between frailty and markers of ongoing clotting processes such as the D dimer and factor XIa-alpha 1 antitrypsin complex. These data are consistent with the hypothesis that frailty is strongly associated with increased inflammation and suggest the potential for a disease-independent inflammatory mechanism for frailty. It is generally accepted that aging is associated with immune dysregulation and with high blood levels of pro-inflammatory immunogenic stimulations. Based on this observation, the term “inflammageing” has been introduced to define a condition that is characterized by elevated levels of blood inflammatory markers and that carries high susceptibility to chronic morbidity, disability, frailty, and premature death in older subjects [181]. “Inflammageing” is a well-recognized risk factor for many diseases that occur at a higher frequency in older patients, including chronic kidney disease, diabetes mellitus, cancer, depression, dementia, and sarcopenia. Although the relationship between “inflammageing” and frailty is well documented, the possibility that interventions could prevent or delay the occurrence of frailty remains to be demonstrated. Additional evidence regarding the relationship between frailty and inflammation was collected during the COVID-19 pandemic. In many studies, the relevant role of overt inflammatory activation in the COVID-19 pathogenesis has been demonstrated. COVID-19 disproportionately affects older people, suggesting that immunosenescence and frailty, often present in older people, may represent a relevant background and could exacerbate acute inflammation that is typically observed in the most severe forms of COVID-19, which is responsible for increased risk of mortality. In this regard, an elevated CRP and a high neutrophil-to-lymphocyte ratio were associated with an increased likelihood of death [182].

### 6.6. Frailty and Aging of the Brain

The subjective perception of aging has also been correlated to the process of brain aging. In this regard, and according to the interoceptive hypothesis [183], the Awareness of Age-Related Changes (AARC) would represent an indirect perception of neurobiological aging [184]. Since the decline in brain volume is associated with age, it is possible, at least in principle, to use the volumetric measurement of brain volume/atrophy to estimate its age. The recent development of new measurement tools has allowed a more precise investigation of the age-related changes in the morphometry of the human cerebral cortex in vivo. Normal aging is associated with changes in volumetric indices of brain atrophy. Whole-brain atrophy can be measured using Magnetic Resonance Imaging (MRI) and brain volume, observed in T1-weighted structural MRIs, may be normalized to the intracranial volume to give an index—the brain-to-intracranial-volume (Brain2ICV) ratio, expressed in percentage. Although the effects of aging are diffuse, some cortical regions do appear to consistently lose volume and/or thickness to a greater extent than other regions during the aging process. The loss of volume and/or thickness primarily affects the prefrontal cortex, particularly the dorsolateral cortex and the dorsomedial prefrontal cortex, but the lateral parietal and lateral temporal association areas are also involved [185,186,187,188,189,190,191,192,193,194]. According to the “last in first out” hypothesis, atrophy first affects the brain regions that develop last, such as the heteromodal association cortices [193,194,195,196,197,198]. In contrast, the “common cause” hypothesis has postulated that a major factor of age-related decline in cognitive performance may be the degradation and loss of structural integrity in components of the nervous system that mediate sensory functioning [199,200,201]. Additional evidence regarding the connection between brain structure and the age-related decline of cognitive abilities, especially memory, is presented in the study performed on older subjects enrolled in the Vallecas Project [202]. This study included 64 subjects (mean age of 81.9 years) classified as “superagers” based on their performances, represented by the delayed verbal episodic memory score, and 55 typical older adults (82.4 years) who scored within one standard deviation of the mean for their age and education level in the same test. During a structural MRI examination, “superagers” exhibited higher grey matter volume cross-sectionally in the medial temporal lobe, cholinergic forebrain, and motor thalamus, and slower total grey matter atrophy, particularly within the medial temporal lobe, compared to typical older adults. A machine learning classification including demographic factors, lifestyle, and clinical predictors showed that faster movement speed and better mental health were the most differentiating factors for “superagers”, suggesting a potential genetic link between them. All these studies indicate the importance of an objective evaluation to assess the “brain age” and more specifically, the relevance of the delta or the gap between the brain’s biological age (estimated) and the chronological age (given) as a marker for the health status of the subject.

### 6.7. Frailty and Nutritional State

Frailty is dependent on energy intake, and in particular, on protein intake (26,203). A deficient energy intake was found to be more frequent among elderly patients, residents of long-term care facilities [203,204], and nursing homes [205]. Reduced protein intake was particularly evident in elderly hospitalized patients [206]. Reduced protein and food energy supply is responsible for evident changes in body composition, measurable using a Dual-energy X-ray Absorptiometry (DEXA) [207,208] or a Bioelectrical impedance analysis (BIA) [209]. Using the BIA, an inverse correlation was observed between phase angle (PhA) and increased all-cause in-hospital mortality in patients admitted to a medical intensive care unit. Therefore, higher PhA values were associated with better cell health, gains in cell function, and better nutritional status, while lower PhA values indicated poor cell health and were associated with frailty, measured via the Korean Modified Barthel Index (KMBI) [210], higher mortality, and mechanical ventilation weaning failure. Therefore, reduced PhA has been suggested as a risk factor for frailty in critically ill patients [211] and in hemodialysis patients [212]. The role of micronutrients in frailty has been evaluated in many studies [213]. The deficiency in micronutrients, such as vitamins B, C, D, and E, and folate and antioxidants, such as the carotenoids and reduced serum levels of trace elements, including selenium and zinc, have all been associated with the development of the multisystemic geriatric disorders responsible for the occurrence of frailty. 

### 6.8. Frailty and Energy Metabolism (the Role of Mitochondria)

The most accepted physiological framework to explain frailty and its consequences was originally proposed by Linda P. Fried and Jeremy Waltson from Baltimore [9,214,215]. According to this framework, frailty is characterized by the dysregulation of inflammatory cytokines and hormones, oxidative stress, malnutrition, mitochondrial dysfunction, sarcopenia, and energy imbalances. Linda P. Fried and colleagues suggested a phenotype based on a clinical vicious cycle of energy dysregulation: the so-called and previously mentioned “frailty cycle” [9,216]. In this scenario, the role of mitochondria is crucial and the connection between energy metabolism, mitochondria, and frailty is sustained by substantial evidence and is found to be a strong factor. During normal mitochondrial respiration, oxygen is partially reduced and gives rise to highly reactive (and unstable) molecules termed “reactive oxygen species” (ROS; mtROS when they are produced by mitochondria). According to the “Free Radical Theory of Ageing” [135], the increase in the free radical’s cellular content, as by-products during normal metabolism, is responsible for the accumulation of damages during the aging process. The demonstration that mitochondria are the main place where free radicals are generated and, concurrently, the main target of free radical action, has led to modifying the original “Free Radical” theory of aging into the “Mitochondrial Free Radical theory of ageing” (MFRTA) [136]. The MFRTA theory was reinforced by the observation that mitochondrial free radical generation (mtROS) is negatively correlated with longevity in mammals [217]. The results of this study, performed in seven different mammalian species, indicated that animals with a longer maximum life span (MLS) produce fewer free radicals than the ones with shorter MLS. This observation was confirmed by another similar study [218]. Considering the direct correlation between the high energy demands of cardiac myocytes and the energy production supplied by the mitochondria, it is not surprising to note that life expectancy is also inversely related to resting heart rate in most mammalian organisms [219,220,221]. All these combined results indicate that when we look at the two extremities of mammalian species in terms of very big or very small animals, we realize that small ones, like mice, have a higher heart rate (310–840 bpm), produce more free radicals and live less (1–3 years), while big mammalians, like whales, show a lower heart rate (33 bpm), produce less free radicals, and live longer (up to 211 years in some species). However, it is important to note that mitochondria play a double role in energy metabolism. On one hand, they are vital because they produce energy for different functions in all cells. On the other hand, they can be toxic; the more energy they produce, the more waste they generate in the form of free radicals. A perfect balance between energy and waste production is therefore necessary to maintain a healthy cell and therefore, a healthy organism. When the cell accelerates its activity, it generates too many waste products that can compromise its function or even its life. With respect to energy production, frailty may be considered a condition of global impairment due to depletion of the physiological reserves. Mitochondria play a central role in this process. They represent the central power unit of every cell in the human body. They are the site of production of the intracellular adenosine triphosphate (ATP), the organic compound that provides energy to drive many processes in every single living cell. However, the importance of mitochondria goes even beyond energy production. Mitochondria can promote immunity by modulating both metabolic and physiologic states in different types of immune cells and for this reason, they are considered central hubs in innate and adaptive immune cells [222]. The role of mitochondria is thus emerging as a major determinant of the immune responses to various infections and diseases [223]. The identification of mitochondria as central players in determining frailty is also based on the analysis of untargeted metabolomics on serum samples of frail and non-frail groups of patients, according to the Rockwood Frailty Index, in Wave 4 of the English Longitudinal Study of Aging (ELSA) [224]. This study identified differences in metabolites and metabolic pathways between these two groups, especially those belonging to the carnitine and vitamin E metabolic pathways. These two pathways are closely related to mitochondrial function. Carnitine stimulates the mitochondrial respiratory rate and is also the molecular carrier through the inner mitochondrial membrane of the long-chain fatty acids (LCFAs) that represent the main substrates for oxidative phosphorylation. Vitamin E is a potent antioxidant and low serum levels of vitamin E, which have been found to be associated with frailty, might contribute to mitochondrial dysfunction by failing to buffer the excess ROS production by the mitochondria. These results indicate that a dysregulated mitochondrial metabolism plays a central role in the induction of age-related frailty [223]. In addition, the quantitation of mtDNA copy number was found to be a strong independent predictor of all-cause mortality in an age- and sex-adjusted race-stratified analysis of 16,401 participants [225]. In a recent systematic review, alterations in the respiratory chain complexes showed significant decrease in activity only in subjects categorized as frail by the Fried Frailty Index [226]. According to these studies, mitochondrial dysfunction has been proposed to play a central role in the aging process and in the acquisition of the frailty phenotype, which includes many aspects, such as genomic instability, telomere attrition, epigenetic alterations, loss of proteostasis, disabled macroautophagy, deregulated nutrient-sensing, cellular senescence, stem cell exhaustion, altered intercellular communication, chronic inflammation, and dysbiosis [227] (Figure 4).

### 6.9. The Accumulation of Morbidities and Frailty

Comorbidity, defined as the concurrence of one or more clinical conditions in a single person, is rather common in elderly persons [228]. A health survey, conducted by the National Center for Health Statistics (NCHS) in the USA, showed that 27.2% of US adults had multiple (≥2) chronic conditions in 2018 [229], and it has been reported that the occurrence of two or more significant conditions increases with advanced age, at a rate of 50% in people under the age of 65 to 62% in those aged 65–74 and 81.5% in those aged ≥ 85 [230]. In this sense, the total number of deficits or diseases seems to be more important than the specific diseases themselves. The ability to respond to such diseases and to accumulating environmental stressors can be reduced when patients are facing preexisting pathological conditions that reduce their physiological energy reserves. In other words, the accumulation of several impairments may be responsible for the development of inefficiency in maintaining homeostasis, and the ability of the body to respond to new and even trivial stressors may be compromised. In this regard, it has been hypothesized that there is a threshold that reaches a certain critical point, after which the vulnerable state may progress in a rapid and nonlinear pattern [231]. According to this hypothesis, frailty can be seen as an expression of the tolerance of each patient’s threshold for the perturbation of their physiological energy reserve. Among the various pathological conditions that people commonly accumulate during aging, particular attention has been given to those that are considered life-threatening, per se. In an epidemiological study performed in California on a sample of more than 4000 patients over 65 years old, the impact of five major pathological conditions, namely hypertension, diabetes, coronary artery disease, cerebrovascular disease, and cancer on frailty and specifically on mortality was evaluated. The study reported a high incidence of all these diseases, with hypertension being the most frequent (57%), followed by diabetes (20%), coronary artery disease (15%), cancer (9%), and cerebrovascular disease (9%). Comorbidity, with the occurrence of two or more conditions in the same patient, was reported in 29% of cases. All five pathological conditions, except hypertension, were predictive of 6-year mortality [232]. The role of comorbidities emerged as an independent factor, capable of affecting mortality during the recent COVID-19 pandemic as well [233]. In the FAIR Health White Paper by West Health Institute and Johns Hopkins University School of Medicine, which involved 467,773 patients diagnosed with COVID-19 (1 April 2020–31 August 2020), it was established that mortality was much higher in those aged 50 and above. Mortality was higher (42.43%) in those over 70 years old, even with the lowest share of COVID-19 diagnoses (4.82%) [234]. At the same time, it was evident that mortality progressively increased with the accumulation of comorbid conditions (Figure 5). In addition, frailty and multimorbidity were associated with the risk of hospitalization with COVID-19 [235].

## 7. Frailty and Diseases

### 7.1. Frailty and Chronic Diseases

Chronic diseases and frailty constitute two major determinants of a person’s health trajectory later in life. The accumulation of chronic diseases is responsible for a decrease in physiological reserves in aging people, thus rendering them more vulnerable to impairment. However, chronic diseases that are ranked as major causes of death or morbidity are usually more prevalent than frailty and it has been reported that only a small minority of people who have multimorbidity, i.e., the presence of several chronic diseases simultaneously, are frail [236]. It is therefore likely that frailty and chronic diseases do not share the same pathogenic mechanisms, suggesting that there is a more complex relationship between them. Frailty could occur as the result of a progression of the chronic disease toward a more advanced stage. The relationship between frailty and chronic diseases has been extensively examined [237]. Since frailty cannot be explained by a linear correlation with single specific chronic diseases or with a combination of these diseases, efforts have been directed to explain its causes using a nonlinear complex adaptive system modeling approach. Frailty can be better explained when we consider the global physiological inefficiencies as the result of impaired homeostasis in several systems instead of looking at single chronic diseases or at derangements in single systems. Such a complex dynamic way of looking at frailty considers the ability of the whole organism to adequately compensate for the derangements caused by one or more chronic diseases that affect single subsystems. According to this model, frailty may emerge when several diseased physiological subsystems fail to react in a coordinated manner and instead, begin to operate semiautonomously, producing compensatory effects that are positive on a smaller scale for the single subsystem but may be dangerous for the entire organism. This situation can be described as the lack of coordination among its subsystems, as a kind of anarchy [238]. Examples of this anarchy are found in all the situations in which there is uncoupling between the benefit to a single subsystem and the benefit to the whole organism. This can be seen in a kidney and its ability to activate hemodynamic, endocrine, and neural compensative mechanisms in response to the reduction in its filtration capacity, which may have deleterious effects at the general level by increasing systemic vascular resistance and left ventricular end-diastolic pressure [239]. The same mechanism can be valid in numerous conditions in which compensative actions, at a certain point, may result in paradoxical negative effects, thus promoting the occurrence of frailty. Another example of this mechanism can be found in the nonthyroidal illness syndrome, in which the reduction of T4 to T3 conversion in peripheral cells offers the advantage of reducing mitochondrial respiration, thus sparing energy in the presence of oxygen reduction but at the same time, it exposes the patient to functional exhaustion. The chronic diseases that have been associated with frailty in published cohort studies of older adults are reported in Table 4 [240]. The prevalence of the reported diseases in women and in patients overall was much more frequent compared to the non-frail patients. Data reported in the cited studies indicate that the prevalence of chronic diseases is often doubled in older frail women compared to non-frail ones. However, there is no specific single disease-frailty association that is markedly stronger than the rest, even if the difference between congestive heart failure and depression in frail compared to non-frail people needs to be considered.

### 7.2. Frailty and Cardiovascular Diseases

There is an intriguing bidirectional correlation between cardiovascular disease and frailty. It has been reported that the prevalence of frailty is higher in people with cardiovascular disease compared to those without the disease [241,242]; however, at the same time, the prevalence of cardiovascular disease has been reported to be higher in people with frailty compared to those who were not classified as frail [243]. Most of the studies reported in the literature are related to the relevant role of frailty as a potential risk factor for the incidence of CVD, while less attention has been given to the role of CVD as a potential risk factor for the incidence of frailty. The relationship between frailty and CVD was initially analyzed in the Cardiovascular Health Study (CHS) performed at the beginning of the year 2000 [244]. The aim of this clinical trial was to identify risk factors that could predict coronary heart disease and stroke in the elderly. This study served as a platform to investigate many potential diseases and risk factors associated with the development of CVD diseases, including a special focus on frailty. Based on these observations, a CHS index was developed by Freid et al. in 2001 [9], which proved to be useful in predicting falls, disability, fractures, and mortality in older subjects [245]. Considering the close association between CVD and frailty, cardiologists strongly recommend routine screening for the presence of frailty and disability in all patients who presented an occurrence of CVD [246]. In addition, frailty was independently associated with the risk of heart failure [247]. In this regard, particular attention to this recommendation has also been given in the most recently published American Heart Association (AHA) guidelines [248].

### 7.3. Frailty and Cancer 

Many studies have analyzed the role and importance of the complex relationship among frailty, sarcopenia, nutritional risk, and malnutrition in patients with cancer. According to these studies, sarcopenia in cancer patients emerged as a major and independent risk factor associated with chemotherapy toxicity, tolerance, and short survival [249,250,251,252]. It is evident that poor nutritional status not only accelerates the progression of cancer, reducing the survival of these patients, but also deeply impacts the tolerability and acceptability of anticancer treatments, with a consequent reduction in the overall quality of life [253]. Malnutrition in cancer patients is the driver of cachexia, a condition characterized by a severe loss of skeletal muscle mass with or without the loss of adipose tissue, the occurrence of systemic inflammation, and the establishment of a negative protein-energy balance [254]. The term “cachexia” originates from the Greek terms “kakos” and “hexis”, meaning “poor physical state”. Different criteria have been proposed to diagnose cachexia in cancer patients [255], as well as in patients affected by cachexia caused by different diseases, including hematological diseases and heart failure [256]. Currently, the widely accepted criteria, proposed by the European Palliative Care Research Collaborative (EPCRC) International consensus, consist of the following: (i) weight loss > 5% over the past 6 months (in the absence of simple starvation), (ii) body mass index (BMI) < 20 and any degree of weight loss > 2%, and (iii) appendicular skeletal muscle index (ASMI) consistent with sarcopenia (male < 7.26 kg/m^2^; female < 5.45 kg/m^2^) and any degree of weight loss > 2% [252]. It has been reported that cancer cachexia can be observed in 50–80% of cancer patients and is responsible for the death of at least 20% of them [257]. Recently, the following parameters used to assess the nutritional state in cancer patients: (i) baseline malnutrition, (ii) Visual Analogue Scale (VAS) score for appetite loss, (iii) albumin < 35 g/L, and (iv) neutrophil/lymphocyte ratio > 3, were independently associated with the death of non-metastatic cancer patients in a retrospective and prospective observational study involving a large number of cancer patients enrolled with the NUTRItional status in the first medical oncology visit ON Clinical Outcomes (NUTRIONCO) study and in the Prevalence of Malnutrition in Oncology (PreMiO) study [258,259]. According to these studies, the presence of malnutrition puts non-metastatic patients at risk of lower survival, similar to metastatic patients. In other words, malnutrition and cachexia can be considered metastasis for people with cancer in terms of their role in predicting outcomes. However, malnutrition and cachexia in cancer patients are still undetected and underestimated in medical practice, and nutritional support is not yet fully managed in accordance with the available guidelines.

### 7.4. Frailty and Neurocognitive System

Neurocognitive disorders with the progressive loss of intellectual abilities, including complex attention, learning and memory, and language and perceptual motor and social cognition are classified as dementia. Such disturbances may be severe enough to affect social and occupational functioning. Cognitive impairment, or cognitive decline, is a major determinant of frailty [260] and is associated with increased mortality and substantially reduces quality of life [261,262,263]. It is interesting to note that a close correlation has been reported between cognitive decline and sarcopenia, creating a link between neurocognitive and physical frailty [264]. Not only do these two conditions appear to be strictly correlated, but they also share common risk factors and pathophysiological pathways. Contracting skeletal muscle is a major source of neurotrophic factors, including the brain-derived neurotrophic factor, and skeletal muscle activity has relevant immune and redox effects that may modify brain function. In this regard, a geriatric clinic in France has implemented a frailty screening tool based on the frailty phenotype by including social and cognitive factors, along with physical components [265]. Recently, similar observations were observed in a study based on a biracial (black and white) cohort of community-dwelling older adults aged 69–79. The subjects included in this study were analyzed for their skeletal muscle content. The correlation between cognitive disorders and physical frailty is indeed relevant when the content of intermuscular adipose tissue (IMAT) is measured via computerized tomography [266]. In this study, the increase in skeletal muscle adiposity predicted declining cognitive function, independent of traditional risk factors for dementia and other body composition depots. The accumulation of fat in the skeletal mass is encountered in the so-called sarcopenic obesity, characterized by high adiposity levels and concurrent low muscle mass and function. This condition is observed in obese and less active patients and becomes more severe with age [267]. Moreover, a close relationship was reported between lower sarcopenia-related indices and cerebral White Matter Hyperintensity (WMH) assessed via magnetic resonance imaging and cognitive impairment, suggesting that the occurrence of WMH may be one of the factors linking sarcopenia and cognitive function [268]. In conclusion, a relevant network exists between the nervous and musculoskeletal systems [269]. The balance among factors produced by the brain, bone, and skeletal muscle is required not only for each tissue homeostasis, but also for the health of other tissues and may be responsible for the occurrence of individual frailty. Both cognitive and physical frailty represent novel targets for the prevention of elderly dependency and diseases. The combined prevalence of frailty and mild cognitive impairment (MCI) was 2.7% among community-dwelling elderly populations [270]. Frailty is a distinct entity measurable in Alzheimer’s disease and MCI, correlating with age and increasing comorbid illness [271]. Physical frailty has also been associated with a greater risk of developing MCI [272]. In two separate studies, the syndrome of frailty and cognitive impairment (“cognitive frailty”, as shown below) has been associated with increased mortality [273,274]. There is increasing research interest in the concept of “cognitive frailty”, i.e., the co-existence of measured frailty and cognitive impairment. Cognitive frailty has been suggested as a significant risk factor for dementia [275] and has been associated with an increased risk of dementia and mortality [276]. Frailty and cognitive impairment have also been found to significantly influence cost/resource utilization by patients [277]. A recent review article has emphasized the importance of assessing frailty and cognitive impairment together [278]. Cognitive frailty is a new term coined by the International Consensus Group (I.A.N.A/I.A.G.G.) that combines physical frailty with cognitive impairment, adding a formal assessment of cognitive function and capturing the psychological aspect of fatigue [279]. A meta-analysis of 24 studies indicated that the prevalence of cognitive frailty among community-dwelling older adults was 9% (95% CI: 8–11%, I^2^ = 99.3%) [280], while another meta-analysis of 14 studies concluded that cognitive frailty better predicts all-cause mortality and dementia than frailty alone [281].

### 7.5. Frailty and Chronic Pain

Chronic pain is one of the most common and costly diseases that affects older adults and contributes to the development of frailty [282]. The association between frailty and chronic pain has been widely observed among older adults, although the hypothesis that they share certain mechanisms is still under investigation [283]. The imbalance between pro- and anti-inflammatory mediators, leading to a pro-inflammatory state, alongside abnormal oxidative stress increase, have been identified as possible common pathways between chronic pain and frailty [284]. Moreover, chronic pain and frailty share the predisposition for the development of cognitive decline and mood disorders [285]. Introduced more than 20 years ago, the concept of “inflammaging”, a condition characterized by a continuous pro-inflammatory state that favors age-related disabilities, could explain both frailty-related progressive decline and chronic pain syndromes [286]. In the last few years, growing evidence has supported the role of neuroinflammation, both in pain chronification processes, including central sensitization [287], and in neurodegenerative disorders, such as Alzheimer’s disease, Parkinson’s disease, and amyotrophic lateral sclerosis [288]. Neuroinflammation is mediated by the release of pro-inflammatory cytokines by non-neuronal cells, such as astrocytes and microglia, which represents the innate immune system of the central nervous system. Neuroinflammation may also involve peripheral immune cells, like mast cells, which, when hyper-activated, may release many pro-inflammatory cytokines and immunomodulators, such as TNF-alpha, IL-6, IL-8, VEGF, histamine, heparin, and tryptase, which play a determinant role in the progression of osteoarthritis (OA) by promoting subcondral erosions, synoviocyte proliferation, angiogenesis, and pain sensitization [289]. Age and frailty are risk factors for the development of OA, which is a leading cause of chronic pain in the elderly [290]. Joint pain is associated with a significant loss of function that affects the frailty status, leading to a vicious circle in which each one affects the progression of the other in a bidirectional way [291]. Chronic pain likely plays a key role as a stressor during aging by worsening functionality and quality of life, which, in the last few years, have become the most relevant outcomes for evaluating the efficacy of analgesic therapies and a key factor for continuing or tapering some pain killers, such as opioids, especially in chronic non-cancer pain [292]. In the Frailty Index, for instance, some health variables are strictly related to chronic pain, such as arthritis, mood disorders, and loss of function in the activities of daily living [293]. Moreover, frailty is often a limiting factor in managing chronic pain syndromes. Older patients may be undertreated due to several reasons. First, there is a common misconception among physicians that the elderly may better tolerate pain compared to younger people, and that pain is the natural consequence of aging. However, the only evidence for this is that the total amount of analgesics required by older people is generally lower than that required by younger ones [294]. Second, the increasing number of comorbidities associated with aging may lead frail patients to be undertreated due to the fear of reduced drug tolerability and subsequent increased incidence of adverse events [295]. Drug-drug interactions, due to the multitude of medications used by the elderly for treating different diseases, may also increase the risk of side effects and discourage physicians from prescribing analgesics. Chronic kidney disease, for example, which is quite frequent at different stages among the elderly, may require special considerations in the selection of analgesic drugs when treating frail patients [296]. Finally, cognitive decline, delirium, and depression may interfere with communication and impair the patient-physician relationship, leading to a difficult pain evaluation in the elderly [297]. The geriatric population is also more likely to undergo surgery and advancing age is an independent factor associated with poor perioperative outcomes [298]. Despite the paucity of frailty-related studies, there is evidence that perioperative inflammatory markers are higher in older people, as a result of a chronic pro-inflammatory status and immunosenescence [299]. These findings may explain why in fragility fractures, morbidity and mortality remain high, and their management requires the identification of specific standards of care [300]. Many pre-operative screening tools have been proposed for the identification of frailty in the surgical population; however, current evidence strongly suggests that these tools are not routinely used [301]. In frail patients, attention should also be paid to the appropriate use of some analgesics that may affect bone health through different mechanisms of action [302], such as opioids, which impair the hypothalamus-pituitary axis [303], and non-steroid anti-inflammatory drugs (NSAID)s, which may affect bone healing after fractures [304].

### 7.6. Frailty and Mental Health 

The World Health Organization defines healthy aging as “the process of developing and maintaining the functional ability that enables well-being in older age” [305]. Opportunities to provide basic needs, move, make decisions, build, and maintain relationships, and be an active part of society are promoted by physical and mental abilities to interact with the environment. These connections are in turn influenced by factors such as family of origin, gender, ethnicity, educational level, and financial resources. Scientific evidence challenges the common view that old age is characterized only by loss and decline, pointing out that it is a risk factor in several pathological conditions but it is also a natural phase of life and an occasion for successful transitions later in life [306,307,308]. The occurrence of frailty is a normal process associated with aging. It is studied as a natural biological event that produces physiological, psychological, and social changes [309], and it is perceived as an inevitable decline associated with old age [310]. The acquisition of a frail phenotype is thus closely associated with the progressive and age-dependent loss of basic molecular/cellular functional properties that cause decreased adaptability to internal/external stress and increased vulnerability to disease and mortality [311]. In this sense, frailty and aging have a common basis in the loss of homeostasis. In aging, the failure in homeodynamics is global whereas in frailty, the failure in homeodynamics is related to energy metabolism and neuromuscular changes [311]. Frailty can be seen as maladaptive aging or “accelerated biological ageing” [312]. It is considered a reflection of biological age, which increases with chronological age and even if it might occur at any age, it is considered a common condition that becomes more prevalent with advancing age. Due to the rise in life span in developed countries, the concept of frailty has been receiving increased public health attention. Life expectancy often does not correspond to healthy life years and all the elderly are at risk of developing frailty; therefore, this concept is increasingly used in primary and specialist care [10,313]. The two most used conceptualizations [314,315,316] are the Frailty Physical Phenotype (FFP) [9] and the Frailty Index (FI) [317]. The former is also referred to as the “physical frailty” approach, while the latter consists of a definition of frailty related to the accumulation of deficits, based on the cumulative effect of age-related medical, functional, and psychosocial disorders. Using the FI, frailty is assumed to be a spectrum of aging and not a syndrome, with a chance to predict death better than the frailty phenotype. However, the FI takes into account disability and may thus not disclose the incidence of disability or future functional impairment as the frailty phenotype. Frailty is more comprehensive than the simple presence or absence of physical deficits or a range of restrictions in activities of daily living. Based on the definition above, age is a determining factor for frailty, but is influenced by several elements, including important psychological factors [314,318,319] and the association with chronic diseases, such as severe mental illness [320]. Current scientific evidence [314,315,316] indicates that frailty is associated with high emotional and psychosocial distress and low perceived social support; in fact, some screening instruments include questions about social activities and loneliness conditions in individuals who are considered potentially frail. Frailty also seems to be related to lower quality of life even in the absence of a major event, such as hospitalization or surgery, and to complications in hospital care, such as longer length of stay and risk of post-surgery deterioration. This condition interacts with disease: It appears that the prognosis is worse in the presence of co-morbidity and polypharmacy. Consequently, the use of medical care increases and the association with premature mortality appears to be established [321]. Frailty is additionally linked to the risk of adverse health outcomes, e.g., falls, fractures, disability, long-term care needs, cognitive impairment, delirium, and depression [322]. Studies converge on the multidimensional nature of frailty: They bring together sensorial, nutritional, and cognitive phenotypes with psychosocial components, including health-related quality of life aspects, reduced motivation, decreased positive mood, and increased vulnerability to emotional stressors [319,321,322,323,324]. Rather than a linear causality model, it seems more appropriate to consider the introduction of a bidirectional relationship to observe how frailty influences the psychological state of the patient and how psychological and social aspects determine the frailty condition. Frailty is also associated with psychiatric disorders and the determination and measurement of frailty is generally considered a valuable clinical assessment in the clinical management of psychiatric patients. Frailty, multimorbidity, and functional impairment are closely associated with psychiatric disorders like psychosis, bipolar disorder, or depression. They are known as a risk factor as well as a consequence of depression in older adults, in patients with psychotic or bipolar disorder multimorbidity, and older adults with cognitive disorders or dementia, resulting in complex interactions of several co-existing diseases [325]. In this regard, individuals with mental disorders are at an increased risk of medical comorbidities, have a lower life expectancy, differ from healthy controls in physiological function, and may experience accelerated biological aging. Frailty represents both a potential mechanism and synergistic factor contributing to the increased mortality risk of individuals with mental disorders [326]. A conclusive definition has not yet been found, but several authors [314,327,328] appear to agree on identifying frailty as a dynamic state resulting from deficits not only in the physical abilities but also in psychological, psychiatric, and social domains. 

#### 7.6.1. Frailty and Stress

Stress is thought to be a key element in the bidirectional relationship between psychological aspects and frailty condition. Stress (eustress and distress) is the physiological or psychological reaction to both external and internal stressors. As reported by the American Psychological Association [329], eustress is a type of stress that results from demanding but sustainable or useful tasks, involves ideal levels of stimulation, and has a beneficial effect on the organism. Distress is a type of stress that results from feeling overburdened by requests, and it is associated with damaging biological modifications and unpleasant emotional states. Such effects, if prolonged due to chronic stress, can result in over-activity of neuroendocrine communication processes and may result in chronic fatigue, hypertension, depression, anger and hostility, acid reflux disease, tunnel vision, migraines, arthritis, sleep deprivation, decreased metabolism, and depotentiation immune system [330,331]. Chronic stress, especially if experienced early in life, can have severe psychobiological consequences, leading to allostatic overload [331,332] and increased oxidative stress [330]. Frequent or prolonged activation of stress responses through the hypothalamic-pituitary-adrenal (HPA)-axis can alter cortisol levels and glucocorticoid signaling and can lead to increased telomere shortening and accelerated aging [330,333], in addition to the dysregulation of the stress response system. Increased HPA axis activity and reactivity can also induce chronic inflammation by activating the immune system and the release of cytokines [330,331]. In the context of frailty, we have a two-way process. Stress can be caused by the frailty condition itself, and thus aggravate it, and at the same time, chronic stress exposure contributes to frailty in the aging process. An interconnection between the effects related to stressful stimuli on the organism and the condition of the frail patient thus appears to be consistent. 

#### 7.6.2. Frailty and Psychosocial Factors

Evidence suggests that frailty is also associated with sociodemographic factors, physical and biological characteristics, lifestyle health, and psychological factors [110,334]. With a particular emphasis on psychological issues, some of the most commonly reported are depressive symptoms experienced by the patient, impaired cognitive function, poor self-rated health, negative affect, poor coping, loneliness, social isolation, living alone, behaviors related to malnutrition, smoking and drinking, polypharmacy, poor sleeping, and poor physical activity. In contrast, protective factors have been identified regarding the frailty condition along with physical exercise, and it seems that cognitive training, psychosocial programs, and depression medication can reduce the severity of frailty. Psychoeducation and psychotherapy interventions along with community awareness of topics related to frailty also seem to be helpful [335]. 

#### 7.6.3. Frailty and Depression

One of the conditions that appear most correlated with frailty is depression. Frailty and depression share many risk factors, consequences, and overlapped criteria [336]. This suggests that there is a bidirectional relationship between frailty and depression, where depressive symptomatology constitutes risk factors for poorer physical health, increased mortality, functional disability in older adults, and consequently, the onset of frailty. On the other hand, the drivers of frailty (including obesity or malnutrition, low physical activity, and smoking) may lead to disability through reduced functioning and subsequently lead to the development of depression [337,338]. Both depression and frailty are associated with a range of deleterious outcomes in older age such as lower quality of life, increased use of healthcare services, and increased morbidity and mortality [338]. Furthermore, co-existing depression and frailty are associated with particularly worse outcomes such as accelerated cognitive impairment and disability [339]. In addition to the main phenotypic descriptions of frailty, which are related to the physical, cognitive, social, and psychological aspects, the depressed frail phenotype has recently been proposed [322]. Indeed, an overlap between late-life depression and frailty has been noted, with symptomatology common to both conditions: psychomotor slowdown, involuntary weight loss, reduced energy, feelings of exhaustion, and decreased activities [340]. These overlapping diagnostic criteria for depressive disorders and physical frailty might partially explain the association between both entities [341] and should be considered to prevent misdiagnosis or inappropriate treatment and to potentially include other frailty models such as the frailty index in the evaluation. The prevalence of depression in a wide sample of people with frailty was 38.60%, while the prevalence of frailty in a large group of people with depression was 40.40%. In addition, individuals with one of the two conditions were more likely to develop the other [339]. Although some studies aim to classify frailty and depression as two different constructs to studying any possible relationships [337,339,340,342], it seems that clinical/subclinical depression is one of the criteria influencing the occurrence of frailty in the patient. Other explanations for this strong and reciprocal relationship between depression and physical frailty are based on the clinical/medical conditions observed in the presence of depression that could contribute to frailty [341]. Depressive disorder is increasingly recognized as a disorder of accelerated aging based on its association with many physiological and cellular markers of aging and, at the same time, frailty is identifiable in depressed patients and prospectively associated with mortality, independently of the extent of multimorbidity, sociodemographic, and lifestyle characteristics [336]. Many aging-related biomarkers are associated with both frailty and depressive disorder, such as neuroinflammation and altered microglia activation in the brain [343] and with increased CRP, interleukin 6 (IL-6), neutrophil gelatinase-associated lipocalin (NGAL), shortened leukocyte telomere length (LTL) [336], lower levels of vitamin D [335], and higher oxidative stress [344,345] in both cerebrospinal fluid (CSF) and blood [346,347]. Recurrent depressive episodes also cause permanent changes in the cortisol response, increasing vulnerability to stressors [348]. Depressed patients with altered cortisol levels show more severe symptoms and poorer responses to treatment [349]. Telomere shrinkage and accelerated cellular aging are also associated with depression and its severity [350,351]. Increased cortisol release observed in depression is thought to have a negative impact on telomeres through the glucocorticoid receptors [350,352]. Depression-induced psychological distress, higher oxidative stress, inflammation, and mitochondrial dysfunction are possible underlying mechanisms in telomere shortening [352]. Further research into the relationship between frailty and depression is essential for a deeper understanding of the underlying mechanisms of these conditions and for developing more effective and holistic strategies for prevention, diagnosis, and treatment.

#### 7.6.4. Frailty and Anxiety 

Although many studies have demonstrated a clear link between frailty and depression, the association between frailty and other mental health conditions is less well-documented and poorly understood. The experience of losing independence and functional decline can trigger anxiety in many elderly adults [353]. Anxiety commonly occurs alongside depression and both share overlapping symptomology (like functional impairment and sleep disturbance), leading to increased risk of disability and frailty. This includes increased use of healthcare services, reductions in functioning, accelerated cognitive decline, and a rise in mortality [354]. Anxiety alone is often less studied in comparison to depression despite having strong impacts on daily life and clear associations with cognitive decline, functional dependence, and increased medical morbidity and mortality [326,354]. The results suggest a potential bidirectional relationship between anxiety and frailty [353]. However, this relationship is complex and can vary among individuals (considering that not everyone with anxiety will experience frailty, and the severity and impact of anxiety can differ significantly), and it may be explained by different factors. First, prolonged or severe anxiety can impact an individual’s overall physical health and contribute to physical frailty, leading to sleep disturbances, poor appetite, and increased stress hormones. Second, people with anxiety disorders may be more prone to adopting a sedentary lifestyle due to their fear or avoidance of certain activities or places. This lack of physical activity can contribute to physical deconditioning, muscle weakness, social isolation, and overall frailty. Third, some individuals with anxiety disorders may turn to unhealthy coping mechanisms, such as smoking, excessive caffeine intake, or overeating, which can negatively affect their physical health and contribute to frailty. Fourth, anxiety often leads to chronic stress, associated with increased inflammation, oxidative stress, and other physiological changes, which can have detrimental effects on the body and can contribute to frailty and physical health problems. Fifth, some medications used to treat anxiety can have side effects that impact physical health, such as falls, cognitive impairment, weight gain, or metabolic disturbances. Finally, anxiety disorders can co-occur with other physical health conditions, such as cardiovascular disease or diabetes, which can increase the risk of frailty and complicate its management [354]. On the contrary, the transition from independence to functional decline and the appearance of physical frailty characteristics (like poor physical performance, decrease in strength, extenuation, and slow gait velocity) can lead to maladaptive psychological responses that include anxiety, stress, feelings of uncertainty and vulnerability, social isolation, and depression [353]. For these reasons, individualized treatment plans and a holistic approach that considers both mental and physical health needs are often recommended for individuals dealing with anxiety and those at risk of frailty. More studies on the evaluation of the associations between frailty and anxiety are recommended. 

#### 7.6.5. Frailty and Suicide 

The association of frailty and disability with suicidal ideation, independent of other depressive symptoms, is an important area of research that has received increasing attention in recent years [338]. Published studies indicate an important interplay and mutually reinforcing process of decline between suicidal ideation, disability burden due to social, interpersonal, and physical limitations and frailty characteristics (including loss of muscle mass, fatigability, and decreased physical capacity) among older adults [335,338]. Several factors contribute to suicidality in frail individuals, including mood disorders, previous suicide attempts, poor physical health, social isolation, and social problems [335]. In the context of depression, there are aspects that are been involved in suicidal ideation, including inflexibility in thinking, rumination, impulsivity, and impairments in cognitive control, and thereby, difficulties in regulating actions and thoughts [335,355]. Furthermore, there are other impairments, like loss of independence, reduced functional capacity, and a diminished sense of utility, value, status, and enjoyment of life that can be significant contributors to suicide risk [335]. However, while most research on suicidal ideation and behavior has traditionally focused on mental health factors (mood disorder diagnosis and past suicide attempts), it is becoming evident that physical health and functional status play a significant role, particularly in older adults [338]. Studies suggest that frailty and both subjective and objective measures of functional impairment (like decreased gait speed and muscle weakness) are significantly related to suicidal ideation, independent of depression severity [338]. Individuals with physical health problems, including chronic illnesses, pain, functional limitations, and disability, may face feelings of hopelessness, frustration, despair, and a desire to end their lives. Also, the impact of these problems on cognitive functioning (such as difficulties in concentration, remembering, problem solving, and communicating with others), social interactions, and quality of life (affecting the ability to perform daily activities, maintain independence, and engage in social and recreational pursuits), has been associated with suicidal ideation, behavior, and attempts in later life [338]. Understanding the association between suicidal ideation and frailty highlights the importance of recognizing frailty, physical weakness, and medical comorbidities as potential indicators for uncovering and addressing suicidal ideation. It emphasizes the need to provide social support and assessment of physical health and functioning as part of suicide risk evaluation for individuals experiencing frailty and disability, especially in older populations [337]. It is thus important to recognize that these individuals may not always exhibit classic signs of mood disorders like severe depression. When increased physical limitations, multiple medical comorbidities, and difficulties with social interactions are reported, investigation of possible suicidal ideation, low mood, or psychiatric symptoms is not only valuable but also a potentially life-saving intervention [338]. Finally, various interventions can help prevent or mitigate both frailty and suicidality in the elderly population, addressing physical, mental, and social aspects, and they include regular exercise, proper and well-balanced nutrition, cognitive training, psychosocial programs, proper medication management, fall prevention, and regular health check-ups [335]. Future studies are needed to explore the reciprocal relationship between the effect of improvement of physical and functional outcomes in suicidal ideation. These studies should encompass a broader range of populations, including younger individuals and those with or without mood-disorder-related symptoms. 

#### 7.6.6. Frailty and Sleep Disorders

Age-related changes in sleep architecture are a natural part of the aging process. These changes often result in lighter and shorter sleep duration in older adults. While these alterations in sleep patterns are considered normal, a significant percentage of older adults experience sleep problems and disorders, such as insomnia (the most common sleep disorder worldwide), restless legs syndrome, periodic limb movements and sleep-disordered breathing [356,357]. Insomnia can have a significant impact on an individual’s daily life and overall health, causing daytime fatigue and sleepiness, muscle tension, headaches, concentration problems, and cognitive impairment. For these reasons, patients with insomnia are more likely to have impaired social, occupational, and daily functioning and other comorbidities, including psychiatric and cardiovascular diseases and other health conditions (diabetes, obesity, chronic pain, etc.) [356,357,358]. An increasing amount of evidence suggests that insomnia is a risk factor for frailty, contributing to the development of multiple somatic and psychological disorders (such as diabetes and metabolic syndrome, hypertension, stroke, coronary artery diseases, anxiety, depression, and cognitive impairment), increasing vulnerability, sarcopenia and increased mortality [356,358,359]. Moreover, some sleep disorders (such as sleep apnea, restless leg syndrome, or periodic limb movement disorder) can contribute to falls by causing nighttime awakenings or leg movements and lead to severe consequences for frail individuals. This can be exacerbated by using benzodiazepines as a treatment for insomnia, which increases the likelihood of falls and confusion and causes further disruption to sleep and leads to dependence problems and withdrawal [360]. Other treatments, such as antipsychotic medication that are prescribed to manage sleep disorders, may also have a negative impact due to their side effects (daytime sedation, increased risk of falls, metabolic disorders, cognitive impairment, etc.). These side effects can exacerbate frailty in older adults as they may lead to reduced physical activity, falls, and functional decline. Poor sleep quality due to sleep disorders may be associated with an increased risk of memory and cognitive impairment by increasing fatigue and reducing cognitive functioning [356,360]. The relationship between poor sleep quality and frailty is evident, and research has shown that it is also reciprocal. While poor sleep quality can contribute to frailty, the reverse is also true: frailty can contribute to sleep disturbances [356]. Individuals with frailty, increased comorbidities, and polypharmacy may experience the development and worsening of insomnia symptoms [361]. For example, frail individuals may experience urinary incontinence or an increased need to urinate during the night, leading to frequent awakenings. In addition, frailty may be associated with respiratory problems, such as reduced lung capacity or muscle weakness, which can impact breathing during sleep. Moreover, age-related frailty and sleep dysfunction often coexist with other comorbidities, particularly psychiatric conditions such as affective disorders (depression and bipolar disorder) and other severe mental disorders like schizophrenia. These shared comorbidities can interact and exacerbate each other, leading to a complex interplay between physical and mental health issues in older adults. Non-pharmacological approaches for managing insomnia in frail individuals focus on improving sleep quality through lifestyle modifications and interventions that address physical, cognitive, and social aspects of well-being, such as physical exercise, nutrition, cognitive training, social participation, and sleep hygiene. These approaches can be especially beneficial for older adults who may be more vulnerable to the adverse effects of medications [356]. More studies are needed for further clarification of the etiological mechanisms and to evaluate whether interventions (pharmacological and non-pharmacological ones) to improve the quality of sleep in older people with insomnia could improve the frailty outcomes.

#### 7.6.7. Frailty and Dementia 

Some review articles have examined the relationship between frailty and dementia [362,363,364]. Two systematic reviews, including one meta-analysis, have examined the relationship between frailty and dementia. Their findings included a pooled prevalence of frailty of 32% in mild to moderate Alzheimer’s disease (AD) and showed that frailty is a significant predictor of all types of dementia [365,366]. According to these studies, frail people were particularly at risk of cognitive impairment and the development of some kind of dementia, including vascular dementia and Alzheimer’s disease compared to non-frail persons [367]. In this regard, frailty has been identified as an independent risk factor for vascular dementia (VaD), independent of all conventional dementia and cardiovascular risk factors [368]. The presence of frailty is thus considered a relevant risk factor for the development of all types of dementia, but especially for the VaD [369]. Wallace and colleagues reported that frailty is associated with an increased likelihood of developing Alzheimer’s disease (AD), independently of AD neuropathology burden [370,371]. Additional evidence regarding the association between frailty and dementia has been found in neuroimaging studies. It has been shown that increasing cognitive decline in frail patients is associated with cortical atrophy, especially in specific cerebral areas [372]. Frailty is also an independent predictor of the incidence of dementia [373]. Higher baseline levels of frailty, as well as more rapid progression of frailty, have also been associated with an increased risk of incidence of AD [374] and death [375]. Frailty indices increase as health deficits accumulate [376], especially those that are related to vascular brain damage [377]. According to a report by the ELSA, not only are frail patients at increased risk of developing dementia, but a similar risk has also been reported in pre-frail patients [378]. Therefore, a thorough assessment of frailty should be included in the evaluation of patients at risk for dementia. In this regard, it has been reported that the coexistence of frailty or pre-frailty and dementia is accompanied by a relevant increase in the burden of care [379,380,381]. Such observations indicate that there is an urgent and increasing need to include the measurement of frailty in the management of patients with dementia [382].

#### 7.6.8. Frailty and Delirium

Delirium is a common condition that affects the hospitalized elderly population. It is a form of acute brain dysfunction characterized by sudden and significant change in an individual’s level of attention and an altered level of consciousness that fluctuates over time. Delirium is typically caused by an underlying medical condition, such as infections, medication side effects, metabolic imbalances, surgery, or other acute illnesses. There are also many predisposing factors such as older age, cognitive and functional impairment, malnutrition, polypharmacy, drug history, and sensory deficits. It is associated with a range of negative outcomes for individuals who experience it, both short-term and long-term, that can be attributed to the acute syndrome of delirium itself, as well as the management of patient care during hospitalization. Delirium can lead to a decline in an individual’s functional status, which often results in reduced mobility, higher dependence, difficulties with activities of daily living (ADLs) and an increased need for assistance with basic tasks. It is also a risk factor for the development of long-term cognitive impairment, which can manifest as dementia or accelerated cognitive decline. It may delay physical and cognitive recovery and can lead to a state of severe frailty at discharge, with a high risk of mortality, institutionalization, and prolonged hospitalization time and an increase in health-related costs [383]. Frailty has been shown to be an independent predictor of residual delirium one year after discharge from a specialist delirium unit [384]. The measurement of frailty using the frailty index (FI) derived from the International Resident Assessment Instrument (interRAI) Comprehensive Geriatric Assessment (CGA) for Acute Care (AC) in more than 1400 patients admitted to several Australian hospitals demonstrated that even a small increase in the Frailty Index score (a 0.1 increase) was significantly associated with many adverse outcomes, including delirium [385]. The literature has consistently reported a higher prevalence of delirium in frail older adults compared to their non-frail counterparts and a large amount of additional evidence indicates that there is a strong association between frailty and delirium [386,387,388,389,390,391]. Frail individuals may have reduced cognitive and physical reserves, making them more susceptible to the physiological stressors that can trigger delirium, such as infections, surgery, or medication side effects and to the cognitive and functional changes associated with this condition. Similarly, delirium can lead to further functional decline in frail individuals, making them more dependent on others and potentially leading to adverse outcomes. In both conditions, sensory impairment (mostly visual and hearing) can play an important role. It can contribute to the development of delirium, as individuals with impaired vision or hearing may struggle to perceive their environment accurately, potentially leading to disorientation and confusion. Additionally, it can contribute to frailty by limiting an individual’s ability to interact with their environment, affecting mobility, balance, and the ability to perform activities of daily living, increasing the risk of falls, accidents, dependency, and isolation. When delirium and frailty were investigated simultaneously, they were found to be associated with increased mortality [392,393]. Both delirium and frailty share certain risk factors (advanced age, chronic medical conditions, cognitive impairment, malnutrition, and polypharmacy), which can contribute to their co-occurrence. The similarities in epidemiological and clinical aspects reported for both delirium and frailty have prompted researchers to suggest the existence of common pathophysiological pathways [394]. However, this topic merits further investigation. Recognizing and addressing the risk factors for both conditions, as well as implementing preventive measures, can help to prevent or minimize the risk of further cognitive and physical declines and improve the overall health and well-being of these individuals.

#### 7.6.9. Frailty and Severe Mental Illness (SMI)

Severe mental illness (SMI) (such as major depressive disorder, schizophrenia, and bipolar disorder) affects around 5% of the adult population. Emerging evidence suggests that people with SMI are associated with multiple medical comorbidities, such as cardiovascular disease and metabolic syndrome, poorer functional status, cognitive deficits, falls, weak muscle strength, and mortality [342,395,396]. People living with severe mental disorders have an average life expectancy that is 15 years lower than those without mental disorders and have an elevated level of frailty [397]. Frailty in the context of severe mental disorders can be defined as a state of vulnerability and physical decline associated with an elevated prevalence of somatic comorbidities (up to twice that of the general population) and premature mortality compared with the general population, mostly through preventable physical causes [326,398]. It is important to note that frailty is typically associated with physical health, but in the context of SMI, it can have significant implications for a person’s overall well-being, representing an important target for prevention and treatment to improve life expectancy and quality. It is likely that mental illness and frailty are mutually reinforcing and may share common risk factors [326], like overweight/obesity, physical inactivity, cardiovascular risk, reduced bone mass, poor self-rated health, and alcohol use [398,399]. The risk of physical comorbidities is significantly increased by severe metabolic adverse effects and interactions of psychopharmacological drugs (such as antipsychotic drugs), polypharmacy, and the comorbidity with substance misuse, which interacts with frailty in relation to risks of weight gain, metabolic and cardiovascular issues, falls, fractures, liver disease, cognitive impairment, depression, and anxiety. In this regard, the symptoms commonly associated with some mental illnesses also have an important impact on functionality, predisposing these individuals to become frail [398]. Positive symptoms, including hallucinations, delusions, and disorganized thinking can affect an individual’s functionality by making it difficult to engage in daily activities, maintain social connections, and seek medical care when needed. Similarly, negative symptoms, such as physical anergia, avolition, and apathy, can significantly impact an individual’s functionality by reducing their ability to engage in self-care, physical activity, and maintain relationships [398]. For example, a lack of motivation may lead to a sedentary lifestyle, poor diet, and neglect of physical health needs. All these factors eventually contribute to physical health problems, obesity, cardiovascular issues, diabetes, and cognitive decline, which are all characteristics of frailty. Those aspects are further influenced by stigma and limited access to healthcare services, high rates of psychosocial stressors, including homelessness, unemployment, loneliness, lack of social support and social instability, and the fact that some people may struggle with maintaining a healthy lifestyle due to their symptoms, limited motivation, or cognitive impairments [397]. This can lead to poor lifestyle choices, including a lack of exercise, poor diet, smoking, and the neglect of their physical health. In addition, recent research has shown that individuals with SMI such as schizophrenia may experience accelerated biological aging [397,398,399,400,401]. This means that people with schizophrenia may exhibit signs of physical aging and related health issues at a faster rate than the general population. The concept of “accelerated biological aging” arose from the evidence that physiological changes in these patients tend to occur earlier than in the general population, and they are associated with early mortality and increased risk of developing certain senescence-associated medical comorbidities including hypertension, heart/cardiovascular diseases, dementia (such as Alzheimer’s disease), Parkinson’s disease, metabolic syndrome, and type 2 diabetes mellitus [401,402]. Several contributing factors have been studied to explain the accelerated aging hypothesis, including an increase in oxidative stress and inflammation biomarkers in patients with schizophrenia [399,401], the association with markers of cellular aging, such as leukocyte telomere length, which suggests that patients with schizophrenia are more likely to age prematurely than the general population [402], the association of schizophrenia with structural and functional brain abnormalities, responsible for cognitive impairment [403], and finally, the use of antipsychotic medications and sedentary and unhealthy lifestyle habits (including poor diet, lack of exercise, and high rates of smoking) that contribute to physical health problems and accelerated aging. Even if there is evidence of accelerated biological aging in individuals with schizophrenia, the exact mechanisms and causative factors are still unknown. The study of these factors can have significant implications for the overall health and well-being of these patients. Similarly, due to the fact that individuals with SMI can become prematurely frail, integrated care and a routine assessment could be used to identify and address frailty in these patients. Thus, it is essential to adopt a holistic approach that considers both their mental and physical health needs. This may involve regular health check-ups, that include monitoring for medication side effects and inappropriate polypharmacy, reducing stress, promoting a healthier lifestyle and behaviors (like physical activity, a balanced diet, and abstaining from addictive substances), improving motivation, enhancing cognitive functioning, providing social support, promoting care continuity and reducing stigma around mental illness to encourage individuals to seek and receive appropriate medical care, and eventually mitigating the impact of frailty and improving life expectancy and quality of life [326]. The individually tailored multicomponent interventions (including physical activity, diet, psycho-social support, and integrated care models) are likely to improve the prognosis for frail individuals with mental health conditions, suggesting that the increased mortality risk associated with frailty and mental disorders can be prevented, treated, and potentially delayed [326]. The concept of “functional recovery” that has become increasingly important between treatment objectives in individuals with SMI involves achieving a productive, autonomous, meaningful, and satisfying life in the community, even with persisting symptoms. The recovery process extends beyond symptomatic remission and includes functional and personal aspects. Addressing factors that contribute to frailty, with a comprehensive and multidimensional approach that considers not only the control of the symptoms but also the physical, psychological, social, and functional factors, is a necessary intervention aimed at achieving full recovery. Further research is needed to explore the biological, psychological, and sociocultural mechanisms underlying the relationship between SMI and frailty, to analyze and prevent the clinical implications of frailty across different psychiatric disorders, including schizophrenia [404], bipolar disorder, and major depressive disorder, and to develop personalized and appropriate interventions to improve quality of life and delay the onset of frailty for this vulnerable population [405].

### 7.7. Frailty and the Endocrine System

The dysregulation of many different hormones has been reported to be associated with frailty. Among them, a reduction in the serum levels of Insulin-like growth factor I (IGF-1) alone [406] or in combination with the reduction of dehydroepiandrosterone sulfate (DHEA-S) [407] is particularly interesting. IGF-I serum levels have been reported to decline during the aging process in many epidemiological studies [408,409,410,411,412,413]. Moreover, reduced IGF-I serum levels are associated with slow walking speed, reduced muscle strength and difficulty with mobility tasks [414]. In addition, low IGF-I confers a high risk for progressive disability and death in older women, especially when associated with high IL-6 levels, suggesting an aggregate effect of dysregulation between endocrine and immune systems [415]. The age-associated hypogonadism that occurs in women after menopause is responsible for the loss of musculotendinous and skeletal quantity and quality. This is considered a natural consequence of the loss of the anabolic effects of estrogens but has clear consequences on health and quality of life and the induction of frailty [416]. The relationship between menopause and frailty has also been confirmed in a scoping review that analyzed a total of 15 studies on this topic [417]. According to this systematic review, early menopause, hysterectomy, low-free testosterone levels, and high C-reactive protein levels are associated with the likelihood of frailty in postmenopausal women [417]. Obesity is a well-recognized condition that is responsible for an increased risk of frailty, as demonstrated in a systematic review and meta-analysis based on 17 studies that were published and performed in community-dwelling older adults [418]. A body mass index (BMI) > 30 kg/m^2^ was strongly linked to a higher risk of frailty and this was more pronounced in patients with abdominal obesity. However, the review reported that an increased risk for frailty was also observed in underweight patients with BMI < 18.5, suggesting that the association between BMI and frailty is a non-linear one, with a “U-shaped” curve [418]. Another condition associated with frailty is osteoporosis, which is defined as increased bone fragility and subsequent increased fracture risk. Frailty is generally considered to be a predictor of osteoporotic fractures [24]. Both the phenotype model [9] and the frailty index of deficit accumulation [419] have been used to show that frailty can be predictive of osteoporotic fractures independent of chronological age in the elderly [420].

#### Frailty and the Low T3 Syndrome

The relationship between frailty and thyroid hormone is rather peculiar. Thyroid hormones are known to be able to regulate the key metabolic pathways that control energy balance, storage and expenditure [421]. They regulate a variety of pathways involved in the metabolism of carbohydrates, lipids, and proteins in several target tissues [422]. It is noteworthy to mention that frail patients admitted to the ICU often show a peculiar hormone condition, known as Low T3 Syndrome or Nonthyroidal Illness Syndrome (NITS). This condition can be observed in up to 70% of patients [423,424,425,426]. Its occurrence is associated with more severe clinical conditions, poor prognosis, and high risk of death [426,427,428,429,430]. Recently, NTIS has been reported to complicate the course of the diseases in patients admitted to the ICU because of COVID-19 [431,432,433,434]. The relationship between thyroid function and COVID-19 was recently examined in a systematic review of studies performed on a total of more than 5800 patients [435]. This systematic review confirmed that thyroid hormone levels, especially FT3 levels, were reduced in patients with COVID-19, compared to the healthy cohort, and this alteration was more evident in severe COVID-19 patients with pneumonia compared to those without pneumonia, indicating that decreased FT3 levels have clinical significance for the prognosis of this condition. The mechanism of this syndrome is not completely elucidated but is generally considered to be an adaptative adjustment to reduced levels of nutrients and oxygen. Hypoxia, particularly at the cellular level, is responsible for the stabilization of the HIF-1-alpha, which may thus activate a cascade of genetic events, including the increase in the transcription of the DIO3 gene [436]. This gene encodes for the inactivating deiodinase type 3, with a consequent reduction in the circulating FT3. Since T3 is one of the major energy stimulator hormones, its reduction is considered to be a measure adopted by the cell to reduce energy use. However, a dramatic side effect of this compensative adaptation is the exhaustion of the cell due to insufficient energy to fulfil its needs for survival. In this context, the occurrence of NTIS in a patient admitted to the ICU for any critical disease, independently of the type and cause of the disease, represents a significant and reliable hormonal marker of frailty. 

## 8. Frailty and Vaccines

Since vulnerable individuals affected by the frailty condition are at increased risk of adverse outcomes, including falls, hospitalization, and mortality, frailty is responsible for an increase in healthcare costs and the resources required to sustain basic needs and to ameliorate the quality of life and social functioning [10]. However, considering that frailty is a dynamic process and that it is preventable, many efforts have been devoted to identifying older adults who are at risk of developing frailty, encouraging lifestyle changes, and promoting specific interventions and preventive actions to recover from frailty. Among the interventions developed to prevent complications, disability, and mortality due to infections in frail subjects, widespread vaccination programs, which are strongly recommended for frail and old patients, have been demonstrated to be successful. A specific vaccination program devoted to frail subjects has been shown to be effective in reducing hospitalization and death rates, as well as healthcare costs. In particular, vaccines against influenza represent the best example of successful preventive action in the 200-year-old history of success against major infectious diseases, which has led to the doctrine of “for each disease, a vaccine” [437]. Other examples include vaccination against poliomyelitis, infections by capsulated bacteria, pertussis, and rotavirus that produce disease through exotoxins, tuberculosis, the coronaviruses. Influenza is one of the major infectious diseases responsible for the increased risk of serious complications, especially during winter, and in populations aged 70 or more [438,439]. Vaccination can be considered the main public health measure available for the primary prevention of communicable diseases and represents the best defense against serious, preventable, and sometimes deadly, complications of infectious diseases. During the 2011–2020 period, in the Global Vaccine Action Plan (GVAP), all countries were called upon to reach the 90% coverage threshold, with all vaccines in the country’s national immunization program by 2020 [440]. The same global approach based on stimulating all States Parties to implement immunization at all ages, has been repurposed in the Immunization Agenda 2030 (IA2030) [441], which builds on the lesson learned from COVID-19, and puts greater emphasis on an immunization strategy tailored to the national context and integrated into primary healthcare services. In this agenda, a strategic priority and a specific commitment were established to extend the benefit of vaccines not only to infants and children, but also to the elderly and most vulnerable people, in other words, the frailest subjects. However, the relationship between frailty and vaccines is a complex and multifaceted one. Frailty and infections are linked bi-directionally, acting both as risk factors and consequences of each other [442]. A well-functioning immune system prevented frailty and vice versa, and adherence to the immunization schedule not only prevented frailty but also maintained the immune homeostasis. However, while vaccines are recommended to reduce the impact of common infections on the clinical conditions of frail subjects who are already compromised by the occurrence of multiple aging-related morbidities, on the other hand, frailty may play a negative role by hampering the response to immunization in older adults, thus reducing the expected protective role of vaccinations. In this regard, several studies suggest that frailty negatively impacts responses to influenza and zoster vaccines in older individuals [443,444]. Lower vaccine effectiveness may be explained in part by immunosenescence, an age-related decline of humoral and cellular immune responses, and “inflammageing”, a state of low-grade inflammation [445,446,447]. In general, two achievements have been crucial for the success of vaccines: the induction of long-lasting immunological memory in individuals and the stimulation of herd immunity that enhances the control of infectious diseases in populations. The use of specific biomarkers of immunologic age, and T cell function, has been proposed to predict protection against infectious diseases and clinical outcomes of vaccination in older adults [447]. In this regard, a lower immunogenicity, consisting of reduced T-cell immune responses to vaccinations, was responsible for a limitation in the clinical efficacy of COVID-19 vaccines administered to nursing home residents [448]. Another aspect of the complex relationship between frailty and vaccination relies on the fact that the vulnerable groups that require vaccines the most, namely the frail subjects, are also the ones that are more susceptible to adverse reactions. According to the results of vaccination programs, including the recent one against COVID-19, older age, geriatric syndromes, and comorbidities have been recognized as major determinants of adverse outcomes [449,450]. All these considerations make it challenging to provide recommendations for vaccination in vulnerable older adults with existing frailty and disability; better consideration should be given to the assessment and management of frail older people and nursing home residents to optimize the best vaccination strategies for such vulnerable subjects. This is also consistent with the position of the WHO that recommends that the State Parties continue to initiate, support, and collaborate on research to gather evidence for COVID-19 prevention and control [451]. In particular, the recommendation was provided to continue to endorse and promote primary research and systematic reviews of research on vaccination efficacy, effectiveness, duration, and safety in groups defined by age, medical conditions, and previous infection and vaccination with various products (Recommendation E).

## 9. Biomarkers of Frailty

With the aim to identify circulating biomarkers of frailty, a systematic approach was recently adopted by the European group, the so-called FRAILOMIC initiative [452]. Using a robust machine learning framework, they were able to identify a limited number of biomarkers involved in frailty predisposition. Three of them were considered protective biomarkers, namely the vitamin D3 (OR: 0.81 [95% CI: 0.68–0.98]), the lutein zeaxanthin (OR: 0.82 [95% CI: 0.70–0.97]), and miRNA125b-5p (OR: 0.73, [95% CI: 0.56–0.97]), while other three were classified as risk biomarkers, namely the pro-BNP (OR: 1.47 [95% CI: 1.27–1.7]), the cardiac troponin T (OR: 1.29 [95% CI: 1.21–1.38]), and the soluble receptor for advanced glycation end products (sRAGE) (OR: 1.26 [95% CI: 1.01–1.57]). It is relevant to note that among these key frailty biomarkers some are related to oxidative stress and the cardiovascular system [452]. Although not included in the above-mentioned study, low peripheral blood levels of Alanine Aminotransferase (ALT) were found to be associated with reduced muscle mass, frailty status, and increased long-term mortality in the older population affected by mielodysplasia [453]. Despite great efforts made to identify biomarkers of frailty, we still lack specific markers other than those associated with the various morbidities that are frequently observed in frail patients but that are not specific to frailty. Some preliminary studies suggest that energy metabolism and mitochondrial function could represent future novel biomarkers for the measurement of the progressive discrepancy between the accumulation of damage and resilience, in other words to assess frailty [224,454].

## 10. Methods to Assess Frailty

Although there is a general agreement concerning the utility of the scales for identifying patients who deserve more intensive and distinctive care plans, there is no consensus on the optimal instruments or the gold standard measurement for validating the diagnosis of frailty in older people. Current available methods differ substantially in their conceptual background and content [455]. According to a systematic review published in 2018, there were 96 studies in which frailty was assessed using 51 different instruments [456]. The various scales used differed in their ability to measure frailty in older adult populations, in the community, as well as in clinical and Long-Term Care Institutions for Older Adults contexts. In addition, they have different abilities to identify the pre-frailty condition and to predict mortality. Such a plethora of frailty assessment tools has been defined as the “wild west of geriatrics”. There is no doubt that frailty scales can be of great help to health managers and physicians to inform decisions on the allocation of resources, especially when they are scarce and during pandemics, such as the recent COVID-19 [457,458,459]. In general, two main approaches can be distinguished in the different approaches applied to measure frailty, namely the unidimensional one, which is focused on measuring physical health, and the multidimensional one, which also includes psychological, social, and more recently, environmental aspects. A partial list of the most popular scales used for frailty assessments is reported in the systematic review by Faller et al. [456], where a total of 51 frailty assessment instruments were considered, with a variable number of included items, ranging from 3 to 92, with different application times ranging from 10 min to several hours, and with different abilities to identify multiple levels of frailty, from only a dichotomous scale (frail versus not frail) to an ordinal scale, allowing the identification of up to six different levels of frailty (robust or not frail, pre-frail, or apparently vulnerable, mild frailty, moderate frailty, and severe frailty). In another study, a total of 35 different methods to measure frailty were analyzed [460], and the various scales considered were subdivided into those that adopted: (i) a phenotype of frailty approach, (ii) a multidimensional approach, (iii) an accumulation of deficit approach, and (iv) a disability approach. According to these criteria, the first group of scales, which adopted a phenotype of frailty approach, included, among others: the Frail Scale (FS), the Physical Frailty Index (PFI), and the Phenotype of Frailty (PHF). The second group of scales, which adopted a multidimensional approach, included the Comprehensive Geriatric Assessment Screening Tests (CGAST), the Edmonton Frail Scale (EFS), the Frailty Staging System (FSS), the Groningen Frailty Indicator (GFI), and the G-8 Geriatric Screening Tool (G8). The third group, which adopted an accumulation of deficit approach, included the Comprehensive Geriatric Assessment (CGA), 40-item Frailty Index (FI40), 70-item Frailty Index Survey of Health, Aging and Retirement in FI70 Europe (FI70), and Long-Term Care Survey Frailty Index (NLTCS). Finally, the fourth group, which adopted a disability approach, included the Hebrew Rehabilitation Center for Aged Vulnerability Index (HRCA), the Canadian Study of Health and Aging Clinical Frailty Scale (SHCFS), the Vulnerable Elders Survey (VE13S) and the World Health Organization Assessment of Functional Capacity and self-reported health (WHRH). Here, we report the principal features of the most common scales. A more comprehensive list of all the scales can be found in the following systematic reviews [456,460,461].

### 10.1. Physical Frailty Phenotype (PFP)

The Physical Frailty Phenotype originates from previous observations regarding the assessment of activities of daily living. The term “activities of daily living” was first coined by Sidney Katz [462]. In the 1984 National Health Interview Supplement on Aging [463], a method to create a scale for physical exercise by interviewing the patients was introduced, also adopted by Duke University as an integral part of its Older Americans Resources and Services Program (the Duke OARS procedures) [464]. These procedures were widely used by clinicians, program evaluators, and planners of geriatric services for a multidimensional functional assessment of older adults. Later on, Fried et al. [9] introduced the concept that a specific phenotype of frailty can be identified by the presence of three or more of the following components: (a) Weight loss (more than 10 lbs), (b) Weakness (grip strength), (c) Exhaustion (self-report), (d) Walking Speed (15 feet), and (e) Physical Activity (Kcals/week) (Figure 6) (Table 5). Patients with 0 points were considered not frail, 1–2 points as Intermediate, and ≥3 points as frail. By using this scale, it is possible to distinguish aging from body composition changes such as loss of muscle mass, reduced muscle quality, and increased fat mass, which altogether precipitate the development of frailty syndrome in older adults. A strong association was reported between the physical frailty phenotype and the risk of developing health-related outcomes. This scale is relatively easy to use, and it is suitable for the rapid assessment of frailty. It has been widely applied in epidemiological studies. In addition to the classical PFP scale, several other scales of physical function have been developed and widely incorporated into clinical practice. Some have been focused on the most basic Activities of Daily Living (ADL), such as personal hygiene or grooming, dressing, toileting, transferring or ambulating, and eating. Others have considered the Intermediate Activities of Daily Living (IADL), including the following items: using the phone, shopping for groceries on their own, planning, heating, and serving their own meals, managing their medicines, cleaning their house or apartment, getting around on their own, either by car, taxi, or public transportation, and managing money and paying bills. Finally, the combination of physical and social functions was considered, introducing the concept of Advanced Activities of Daily Living (AADL). Basic ADL scales have been shown to be valuable in predicting in-hospital mortality and discharge to nursing homes [465], mobility, and recovery from conditions such as hip fracture [466]. Intermediate Activities of Daily Living scales are frequently used to decide whether an older person can continue to live alone independently. AADL, which takes into account physical and social functions that tend to be voluntary, represents a better and precocious predictor of functional decline compared to ADL and IADL [467]. This scale proved to be useful in evaluating not only normal cognitive aging, but also MCI and dementia [468].

### 10.2. The Clinical Frailty Scale (CFS)

The CFS was introduced in 2005 by the second clinical examination of the Canadian Study of Health and Aging (CSHA) to summarize the overall level of fitness or frailty of an older adult after they had been evaluated by an experienced clinician [419]. Although it was introduced as a means of summarizing a multidimensional assessment in an epidemiological setting, the CFS quickly evolved for clinical use, and has been widely taken up as a judgement-based tool to screen for frailty and to broadly stratify degrees of fitness and frailty. It is not a questionnaire, but a way to summarize information from a clinical encounter with an older person, in a context that is useful for screening for and roughly quantifying an individual’s overall health status. The highest grade of the CFS (level 7), as published in 2005, incorporated both severe frailty and terminal illness. Later, it became evident that the severely frail, very severely frail, and terminally ill should have been considered clinically distinct groups who deserved distinctive care plans. Therefore, the CFS was expanded from a 7-point scale to the new 9-point scale [469]. Another change was made in 2020, and the revised version (2.0) of CFS included a modification to level 2, which changed from “Well” to “Fit”, and level 4 was modified from “Vulnerable” to “living with Very Mild Frailty”. Levels 5–8 were restated as “living with mild frailty”, “living with moderate frailty”, “living with severe frailty”, and “living with very severe frailty”, respectively [459]. The CFS score was found to be predictive of death in patients admitted to the intensive care unit for COVID-19 in an observational cohort in a Swedish study, with higher scores associated with increased mortality [470] (Table 6).

### 10.3. The Cardiovascular Health Study (CHS) Criteria 

The Cardiovascular Health Study (CHS) criteria, based on a phenotype model, represent one of the most popular measures for diagnosing physical frailty. It uses objective and relatively easily measured criteria and has been shown to detect a population of patients with a high risk of falls, disability, and death [9]. There is also a Japanese version [471], which was revised in 2020 [472]. It aimed to identify risk factors for cardiovascular disease in adults Americans aged 65 or older [9]. The assessment of frailty in cardiac patients is most relevant [473]. Frailty can worsen prognosis in such patients, especially in those undergoing cardiac surgery and other cardiovascular interventions and may reduce the net benefits of these cardiac interventions [474]. This method is widely used in multiple epidemiological studies, with a good predictive value for adverse clinical outcomes and can also be used for the assessment of other disorders, including pulmonary disorders, diabetes, kidney disease, and vascular dementia. 

### 10.4. The Edmonton Frail Scale (EFS)

The Edmonton Frail Scale (EFS) is an index used to measure alterations related to frailty by assessing nine subscales, namely cognition, general health status, functional independence, social support, medication use, nutrition, mood, continence, and functional performance (expressed in a total of 17 points maximum) [475]. According to the classification of frailty, based on this index, 0 to 5 points indicate the absence of frailty, 6 to 7 indicate vulnerability, 8 to 9 indicate mild frailty, 10 to 11 indicate moderate frailty, and scores exceeding 12 indicate severe frailty [475,476] (Table 7).

### 10.5. The Five-Item Frailty Screening Index 

The 5-item FRAIL Score is an index comprising five categories which correspond to: fatigue, resistance, ambulation, illnesses, and loss of weight (Table 8). Fatigue was determined by inquiring about the individual’s feeling of exhaustion; resistance was evaluated according to the patient’s ability to climb stairs; ambulation was considered when the individual was able to walk; illnesses corresponded to the presence of at least 5 pre-defined illnesses, and weight loss was determined if the individual had experienced a weight reduction of 5% in the last 12 months [477].

### 10.6. The Frailty Index (FI) 

The Frailty Index (FI) is a measure of the cumulative proportions of up to 30 of the more common diseases or causes of ill health (“deficits”) [133]. The frailty index (FI) is one of the leading paradigms for understanding frailty in populations at different levels, including personal, organ, and tissue [478,479].

### 10.7. The Comprehensive Geriatric Assessment (CGA)

The Comprehensive Geriatric Assessment (CGA) is commonly used in geriatric medicine to collect relevant information regarding the health status and function of older patients and is mainly aimed at managing patients in a multidisciplinary way, rather than simply assessing the diagnosis [480]. A careful CGA, especially when used in association with the frailty index to construct a standardized CGA, allows to collect relevant past and current information regarding a patient’s activities of daily living, mobility, and cognition, as well as facts regarding personal medical history, including chronic and acute diseases [479,481].

## 11. The European Guidelines

The European Commission (EC) has prioritized frailty within the health policy agenda of the majority of the European Union (EU) member states through its “Joint Action on Frailty Prevention” (ADVANTAGE Joint Action) consortium [482]. The ADVANTAGE Joint Action was aimed at building a shared understanding among policymakers and stakeholders to develop a common European approach to frailty prevention and to allow the definition of the specific roles of professionals involved in the process of comprehensive geriatric assessment for different settings [483]. Such approaches will provide the basis for building a consensus-based European multi-professional capability framework for frailty prevention and management. Routine screening for frailty in older adults has been strongly recommended by national and international guidelines [18,484]. ADVANTAGE is a Joint Action (JA), co-founded by the European Commission under the third European Union (EU) Health Program 2014–2020, involving 22 Member States and 35 organizations [482]. Partners have worked together to summarize the current State of the Art of the different components of frailty and its management, both at a personal and population level, and increased knowledge in the field of frailty in order to build a common understanding of frailty to be used by the Member States. The final output of the project was planned to be the “Frailty prevention approach”, a common European model to tackle frailty, indicating aspects that need to be prioritized in the upcoming years at the European, national, and regional levels and develop a common management approach of older people who are frail or at risk of developing frailty in the EU [482].

## 12. Interventions to Reduce Frailty

Despite the profusion of studies concerning the relevance of frailty in morbidity and mortality, especially in older subjects, frailty is seldom measured and factored into clinical decision-making, evaluation of the prognosis of at-risk older adults, and establishment of timely and adequate preventive measures [485]. Therefore, evidence for demonstrating that interventions designed to improve frailty may lead to better outcomes in elderly patients is currently limited. In the consensus statement published in 2013, four possible areas of interventions that could lead to some improvements in frailty have been indicated [18]: (i) exercise (resistance and aerobic), (ii) caloric and protein support, (iii) vitamin D supplementation, and (iv) reduction of polypharmacy. 

### 12.1. Vitamin D Supplementation 

Vitamin D deficiency in older persons is rather frequent and it has been estimated that there is an age-related reduction in vitamin D production of 13% per decade, with a 50% reduction in production at 70 years compared to that at 20 years [486]. Its supplementation has been shown to reduce falls [487], hip fractures [488], and mortality [489]. In addition, vitamin D was found to be reduced in COVID-19 patients, affecting the prognosis of these patients [490], and its supplementation in frail elderly COVID-19 patients, administered in regular bolus during the disease, was effective at reducing the severity of the clinical outcome and increasing the survival rate [491].

### 12.2. Weight Loss

Weight loss, undernutrition, and sarcopenia are well-known clinical components of the frail syndrome [492,493]. A large amount of evidence indicates that appropriate calorie supplementation, especially from proteins, may be effective at increasing weight, muscle mass, and muscle strength [494,495,496].

### 12.3. Polypharmacy Reduction

The relationship between polypharmacy is complex because of the confounding effect of comorbidity and frailty. However, polypharmacy, especially excessive polypharmacy with more than 10 medications, is a relevant determinant factor of frailty [497] and is associated with an increased risk of mortality [498]. 

### 12.4. Increase in Physical Activity

Among all the possible interventions to manage frailty in elderly people, physical exercise proved to be most effective in inducing the best and consistent benefit [499,500,501]. Several studies have reported the benefit of exercise in reducing hospitalization and nursing home placement in the course of many different diseases, including hip fractures [502] and in the rehabilitation process after several cardiovascular diseases, such as acute myocardial infarction (MI), stable angina, heart failure, cardiac transplant, or after major procedures such as percutaneous coronary intervention (PCI), coronary artery bypass graft (CABG), or trans-cutaneous aortic valve replacement (TAVR) [503,504,505]. 

It is surprising to note that despite improvements in outcomes observed when such interventions are prescribed, this facility remains underutilized in current clinical practice. A careful and simple frailty screening should be routinely performed, especially in people aged 70 years or more, to identify frail subjects and to prescribe appropriate interventions that have been shown to produce more beneficial than harmful outcomes [506]. In this regard, a recently agreed Pan-European multi-professional capability framework for frailty prevention and management was developed with the support of the JA ADVANTAGE and the European Geriatric Medicine Society (EuGMS) to promote the recognition of frailty, furthering advancements in evidence-based treatment options, and to identify cost-effective care delivery strategies [507].

## 13. Frailty and Implications for Policy and Practice

The recognition of the frailty syndrome among older adults is gaining attention as an emerging public health priority [508]. Frailty could represent a suitable criterion for risk stratification in older adults, who are the main users of medical and social care services. Most of the National Health Systems in Europe are not well prepared to deal with the chronic and complex medical needs of frail older patients, especially when they are facing an increase in the aging of the older population driven by the rise in their life expectancies [509]. According to a Eurostat projection, the population of older people (defined here as those aged 65 years or more) in the EU-27 will increase significantly, rising from 90.5 million at the start of 2019 to 129.8 million by 2050. During this period, the number of people in the EU-27 aged 75–84 is projected to expand by 56.1%, while the number of people aged 65–74 is projected to increase by 16.6%. In contrast, the latest projections suggest that there will be 13.5% fewer people aged less than 55 years living in the EU-27 by 2050. However, we should consider that the group of older adults is a highly heterogeneous one, with variable health conditions. The complex interactions of genetic, biological, and environmental backgrounds, as well as other physical, psychological, and social factors are the reasons why the chronological age may be more different from the biological age in these subjects [510]. In addition, even if it is widely accepted that the mean prevalence of frailty gradually increases with age, many longitudinal population-based studies indicate that the biological age, and therefore frailty, is not unpreventable and may even improve with age spontaneously [511] and in response to appropriate interventions and adequate lifestyle changes [512,513,514,515,516,517,518] and that some of them had frequent and dynamic transitions over time [104]. These observations highlighted the importance of measures aimed at improving the quality of care for frail older adults. The implementation of measures devoted to modifying the lifestyle, ameliorating nutritional behavior, and promoting cognitive health maintenance would promote healthy aging, thus diminishing the impact of frailty on healthcare systems and strengthening their sustainability [519,520]. Frailty assessment, prevention, and management thus represent major aims for health and social care professionals, scientists, and public health experts to provide adequate health action planning policies. Many experts and scientific associations advocate a major change in the management of frailty that would be directed towards individually tailored interventions and aimed at preserving an individual’s independence, physical function, cognition, and social activity [520].

## 14. Frailty and Digital Health

In order to obtain a reliable determination of older adults’ frailty state and to plan a properly targeted intervention aimed at quality of care, it appears crucial to have appropriate, individual, and timely measurements of physical parameters of frailty. The gait speed, as well as other quantitative parameters of gait, including baseline stride length, stride variability, stride width, stride velocity, and step width are considered useful parameters to predict not only falls but also major clinical adverse events in frail older adults, as reported in an extensive and systematic review and meta-analysis [521]. In this regard, the use of body-worn inertial sensors to enhance the accuracy of standard clinical mobility assessments has been proposed to measure gait speed and parameters more accurately. In a study, the use of a touchscreen mobile assessment platform using wireless inertial sensors (and pressure sensors for the balance test) was applied to quantify the balance and mobility of older adults during the Timed Up and Go (TUG) test, the five-times-sit-to-stand test (FTSS) and balance assessments [522]. To this purpose, Inertial sensors (SHIMMER, Shimmer research, Dublin, Ireland), containing tri-axial accelerometers and a tri-axial gyroscope, were used to quantify movement during each assessment and data obtained were used to obtain a better assessment of patients’ risk of falls and frailty, which can be used for a more individualized treatment plan. In particular, these measurements, in combination with the classical frailty scales, may offer a more clinically useful measure of a patient’s physical state than either measure alone. The use of digital electronic devices has been reported to have positive impacts on cognitive function and well-being after educating older adults about digital technology [523]. Although digital health interventions using smartphones, wearable devices, and other devices are still expensive [524], similar mobile applications could be used as one of the social prescriptions for measuring, preventing, and improving frailty and sarcopenia [525]. Another digital approach has been applied in a recent nationwide multicentric study performed in seven Italian regions [526]. In this study, all the various factors that influence an individual’s state of health towards frailty, namely the environmental, medical, educational, economic, and psychological factors, were collected and digitally analyzed using an information technology-supported multidimensional approach [527] to detect the frailty risk factors at a very early stage. Such measurements are also important to obtain objective and individual parameters for a more personalized assessment of frailty. The European project SUNFRAIL proposes an innovative and integrated approach that takes into account all the various aspects of frailty and is aimed at identifying and implementing innovative approaches for the prevention and management of frailty, integrating the “biomedical” paradigm of frailty [15] with the “biopsychosocial” paradigm [528]. 

## 15. Concluding Remarks and Future Perspectives

In conclusion, the occurrence of frailty in elderly people represents a major issue in many countries that are facing a serious demographic challenge related to the aging of their citizens. In particular, the recent COVID-19 pandemic has imposed health challenges of unprecedented dimensions, and humanity has, once again, faced the deadly impact of infectious diseases, spreading in waves, impacting the frail and older subjects more severely. The question regarding who should be considered frail is of particular interest both for medical reasons and for planning and politics of resource allocation for primary health care. The correct recognition and adequate management of frailty thus represents a major challenge in geriatric assessment. According to the extensive literature on this topic, it is clear that the frail syndrome is very complex, involving many domains and affecting multiple physiologic systems. Its management should be directed toward a comprehensive and multifaceted holistic approach, involving many interrelated fields. Adequate recognition and management of frailty with such a multidomain and personalized intervention strategy may slow down its progression, and since frailty is fully preventable, it may even be able to reverse the course of this condition completely. It could be possible to improve the clinical outcomes of these subjects, thus leading to dramatic improvement in the sustainability of health and social systems too. Therefore, as indicated by the Joint Action ADVANTAGE, co-founded by the European Commission, there is an urgent need to increase knowledge in the field of frailty. A call to action is necessary to promote remarkable changes in the organization and implementation of the health and social systems and to plan a common “frailty prevention approach” for successful aging or aging without disability or loss of independence. Intensive research investigations should be directed toward the identification of new accurate and individual biomarkers for a more personalized frailty assessment, both among community people and hospitalized patients. In this sense, the assessment of mitochondria integrity and function in frail patients represents a promising area of investigation [529]. Mitochondria are the energy power unit of the body, and they are at the crossroads where many pathogenic mechanisms converge. It is thus likely that in the future, mitochondrial respiration activity will be acknowledged as the new biomarker that we have been searching for a long time and it will be included among the various methods routinely used to assess frailty, as a novel putative target for new types of personalized interventions to prevent and treat frailty. 

## Figures and Tables

**Figure 1 jcm-13-00721-f001:**
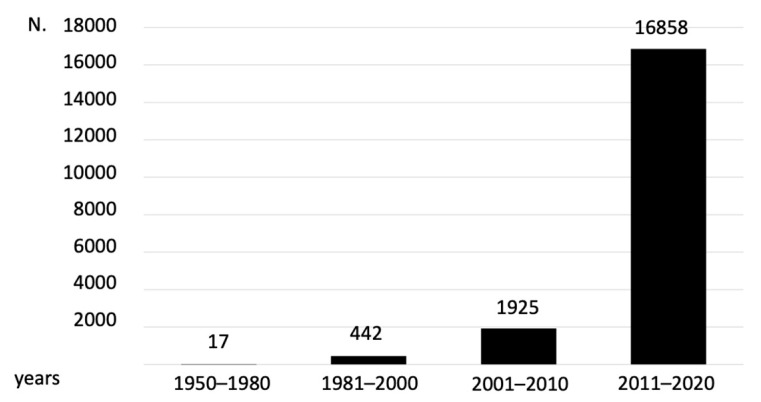
The number of papers published in the literature citing the term “frailty” and published since the year 1950 are reported for the indicated years. Data were obtained from Pubmed, the National Library of Medicine, and the U.S. Department of Health and Human Services (HHS), available at https://pubmed.ncbi.nlm.nih.gov (accessed on 2 November 2023).

**Figure 2 jcm-13-00721-f002:**
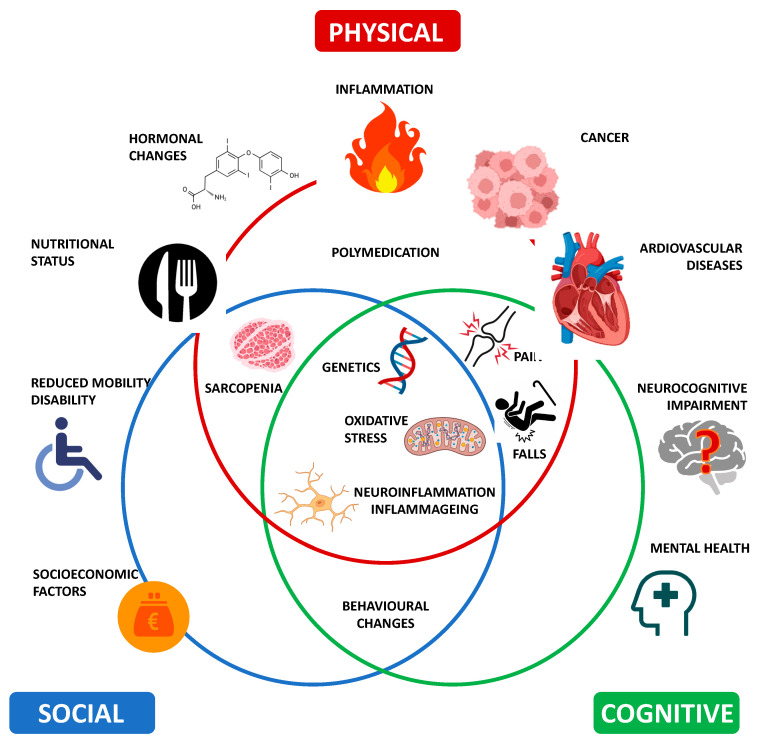
The complex network of frailty involving different domains. The different factors that influence frailty are reported.

**Figure 3 jcm-13-00721-f003:**
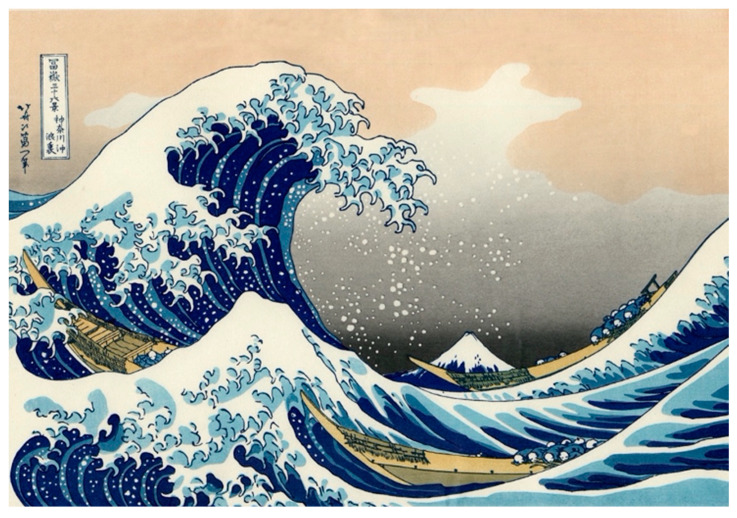
The Great Wave off the Coast of Kanagawa. Katsushika Hokusai, Japan 1831 [33].

**Figure 4 jcm-13-00721-f004:**
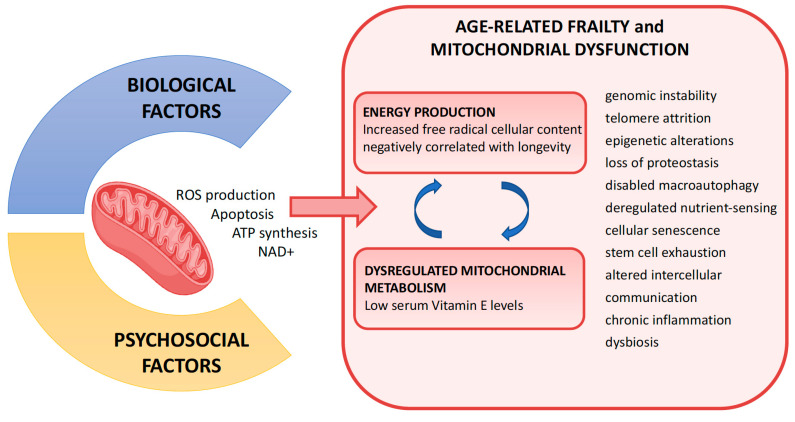
The central role of mitochondria in aging.

**Figure 5 jcm-13-00721-f005:**
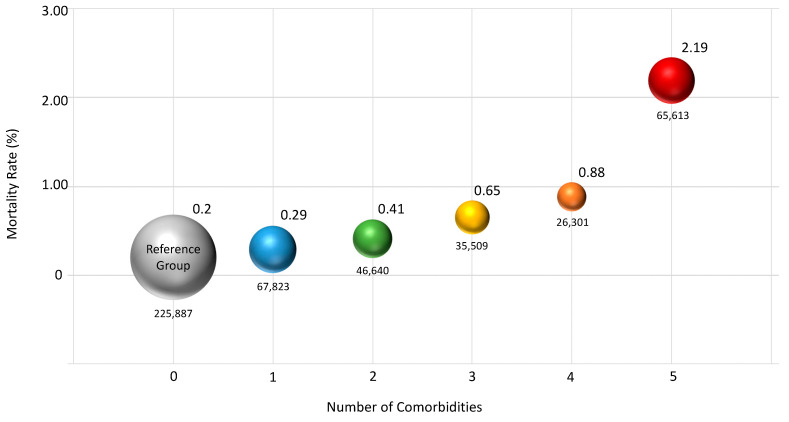
Risk factors for COVID-19 mortality. Adapted from [234]. gray circle, reference group; blue circle, 1 morbidity; green circle, 2 morbidities; yellow circle, 3 morbidities; orange circle, 4 morbidities; red circle, 5 morbidities.

**Figure 6 jcm-13-00721-f006:**
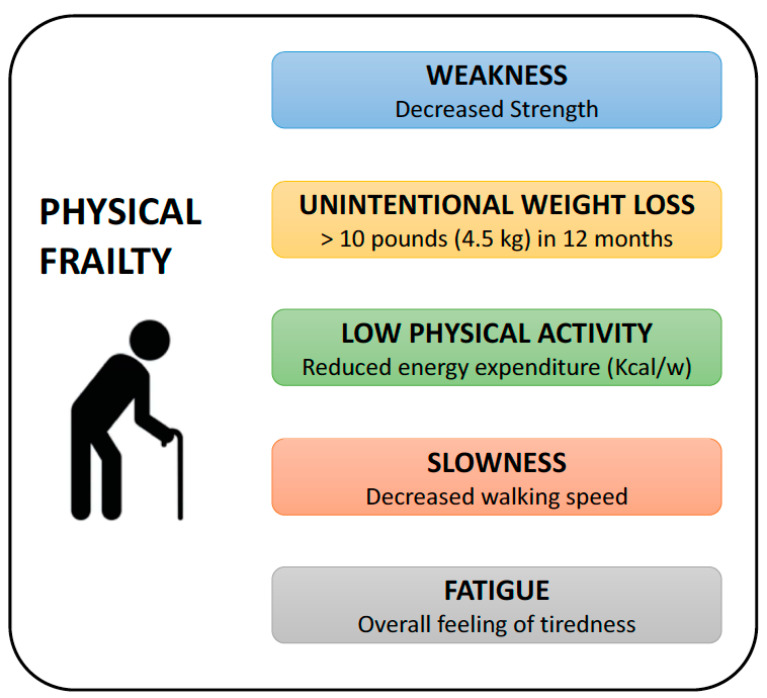
Physical frailty based on the Fried phenotype model. Adapted from Fried et al. [9].

**Table 1 jcm-13-00721-t001:** Frailty definitions in different areas and domains.

Physical/Lifestyle	Psychological	Sociodemographic	Economic	Educational	Jurisprudential
Clinical	Psychological	Social	Economic	Educational	Fragilitas sexus
Physical	Cognitive	Migration	Global	Health Literacy	
Skeletal	Perceived				
Neurological					
Endocrine					
Oral					
Tendon					
Joint					
Renal					
Skin					
Handgrip Strength					
Sensorial					
Sedentary Behavior					

**Table 2 jcm-13-00721-t002:** Prevalence of frailty among community-dwelling older adults (**A**) and in nursing homes (**B**).

(**A**)			
**Pooled Estimates of Incidence**	**Pre-Frail People**	**Frail People**	**Reference**
Community-dwelling older adults	150.64 cases per 1000 person-years (95% CI = 123.26–184.08) (21 studies, *n* = 1163)	43.36 cases per 1000 person-years (95% CI = 37.29–50.41) (46 studies, *n* = 1163)	[100]
(**B**)			
**Pooled estimates of prevalence**	**Pre-frail People**	**Frail People**	**Reference**
Nursing homes	52.3% (95% CI = 37.9%–66.5%) (9 studies, *n* = 1373)	40.2% (CI = 28.9%–52.1%) (7 studies, *n* = 1163)	[101]

**Table 3 jcm-13-00721-t003:** Prevalence of frailty in Europe.

Pooled Estimates of Prevalence in Europe (22 Countries)	Frail People	Reference
Overall	18% (95% CI, 15–21) (62 studies, representing 68 unique datasets, *n* = 1163)	[106]
Community	12% (95% CI = 10–15) (53 datasets, *n* = 1163)	[106]
Non-community	45% (95% CI = 27–63) (15 datasets, *n* = 1163)	[106]

**Table 4 jcm-13-00721-t004:** Frailty and chronic diseases. Adapted from [240]. References for the single studies can be found in [240].

Chronic Disease	Overall Prevalence in Frail patients (%)	Prevalence in Frail Women (%)
Hypertension	50.8–53.1	60.8
Chronic kidney disease	-	54.3
Osteoarthritis	25.9–70.8	78.2
Depressive symptoms	-	46.3
Coronary heart disease	-	17.2–41.5
Diabetes mellitus	13.6–25.0	9.9–21.3
Chronic lower respiratory tract diseases	12.3–14.1	9.8–15.5
Myocardial infarction alone	8.6–13.3	-
Rheumatoid arthritis	-	6.4
Stroke	12.3	4.4
Peripheral arterial disease	3.8–14.8	-
Congestive heart failure	12.3–13.6	3.5

**Table 5 jcm-13-00721-t005:** Physical frailty based on the Fried phenotype model. Adapted from Fried et al. [9].

Frailty Criterion	Definition
Unintentional Weight loss	Loss of more than 5% body weight unintentionally in the last year or BMI is less than 18.5 kg/m^2^
Exhaustion	Feeling unusually tired/weak (by rating the usual energy level on a self-reported scale from 0 to 10)
Slowness	Walking speed measured in a 4-m length in m/s
Low Activity Level	Physical expenditure measured in Kcal/week
Weakness	Grip strength measured using a hand dynamometer and by squeezing the dynamometer maximally 3 times with the dominant hand

**Table 6 jcm-13-00721-t006:** Clinical Frailty Scale (CFS). Adapted from Rockwood et al. [419].

#	Frailty Categories	Definition
1	Very fit	People who are robust, active, energetic, and motivated. These people exercise regularly and are among the fittest for their age.
2	Well	People who have no active disease symptoms but are less fit than the “very fit” category. They exercise or are very active occasionally, e.g., seasonally.
3	Well	People whose medical problems are well-controlled but are not regularly active beyond routine walking.
4	Apparently vulnerable	While not dependent on others for daily help, their symptoms often limit activities. A common complaint is being “slowed up” and/or being tired during the day.
5	Mildly frail	These people often have more evident symptoms of slowing and need help in high order IADLs (finances, transportation, heavy housework, and medications). Typically, mild frailty progressively impairs shopping, outdoor walks on their own, meal preparation, and housework.
6	Moderately frail	People need help with all outdoor activities and housekeeping. Indoors, they often have problems with stairs, need help with bathing, and might need minimal assistance (cuing and standby) with dressing.
7	Severely frail	They are completely dependent for personal care, regardless of the cause (physical or cognitive). Even so, they seem stable and not at high risk of dying (within ~ 6 months).
8	Very severely frail	They are completely dependent, approaching the end of life. Typically, they may not even recover from a minor illness.
9	Terminally ill	They are approaching the end of life. This category applies to people with a short life expectancy.

**Table 7 jcm-13-00721-t007:** Edmonton Frail Scale (EFS). Adapted from Rolfson et al. [475].

#	Frailty Categories	Definition	0 Points	1 Point	2 Points
1	Cognition	Based on the Clock Drawing Test (Placing numbers in the correct positions on a pre-drawn circle and placing hands to indicate the time of “ten after eleven”	No Errors	Minor spacing errors	Other errors
2	General Health Status	Previous admittances to a hospital Self-description of health	0 time Excellent, Very good, or Good	1–2 times Fair	>2 times Poor
3	Functional Independence	Help needed for daily activities such as: meal preparation, shopping, transportation, telephone, housekeeping, laundry, managing money, and taking medications	0–1	2–4	5–8
4	Social Support	Reliance on someone who is willing and able to meet their needs	Always	Sometimes	Never
5	Medication Use	Assumption of five or more different medications on a regular basis They forget to take the medicine at the right time	No No	Yes Yes	
6	Nutrition	Weight is lost such that the clothing has become looser	No	Yes	
7	Mood	Feeling sad or depressed	No	Yes	
8	Continence	Losing urine control	No	Yes	
9	Function Performance (balance and mobility)	Based on the Timed Up and Go test (They will sit in a chair with their back and arms resting and then stand up on the command “GO” and walk at a safe and comfortable pace (approximately 3 m away) before finally returning to the chair and sitting down)	0–10 s	11–20 s	>20 s unwilling needs Assistance
	Totals		/17		

Score description: 0–5 = Not Frail, 6–7 = Vulnerable, 8–9 = Mild Frailty, 10–11 = Moderate Frailty, and 12–17 = Severe Frailty.

**Table 8 jcm-13-00721-t008:** The 5-item FRAIL Score.

#	Frailty Categories	Definition
1	Fatigue	Too tired to exercise
2	Resistance	Able to climb a flight of stairs without assistance
3	Ambulation	Able to walk one block without assistance
4	Illnesses	Presence of five or more illnesses (Heart attack, angina, heart failure, stroke, dementia, COPD, diabetes, malignancy, osteoarthritis, hypertension, asthma, and kidney disease)
5	Loss of weight	Loss of more than 5% of body weight in one year

## Data Availability

No new data were created or analyzed in this study. Data sharing is not applicable to this article.

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
