# Peer review of "To Be Frail or Not to Be Frail: This Is the Question—A Critical Narrative Review of Frailty"

_jcm, 2024, doi:10.3390/jcm13030721_

Round 1
Reviewer 1 Report
Comments and Suggestions for Authors
1. This is a VERY long review and it is more appropriate as a book. I recommend shortening so more will actually read it.
2. I recommend adding notes regarding the way of measuring and diagnosing sarcopenia and frailty by LOW ALT blood values
Author Response
1. We thank reviewer #1 for his/her appreciation to our work. We recognize that the manuscript may be considered too long and we have reduced the length. In particular, we substantially reduced the paragraph of Mental Health (more than 5% of the words count). Removed part of the review have been marked and modified text has been highlighted in the version with corrections.
2. Thank you for your suggestion. We have included a note regarding sarcopenia and low ALT serum values in the paragraph of “Biomarkers of Frailty” and a new reference has been included in the reference list (Ref. # 454). The numbers of the references have been modified accordingly.
3. A new affiliation has been added to the first Author.
Reviewer 2 Report
Comments and Suggestions for Authors
The paper is very extensive, but the historical revision of frailty in pandemics is oversize; currently, narrative reviews are only valid when they are written by highly prestigious professionals. If not, it is the format of a systematic review, with meta-analysis if possible, that provides useful and relevant information for clinical practice. This paper doesnt match with the objective of JCM to publish scientific and relevant data and reserachs.
Author Response
- We thank the reviewer for the suggestion. We have considered the possibility to perform a meta-analysis but we believe that such approach would have not been representative of the complex nature of frailty. There are so many aspects of frailty and so many different ways to measure it, as well as so many areas and domains where the term frailty has been used to define specific conditions of vulnerability that we think almost impossible to constrain and combines all the results of the numerous studies published on this argument, into a single statistical analysis, based on strict protocols such as the PRISMA protocol. We believe that the traditional narrative review, based on the critical overview of previously research published in the literature, would represent better the complexity of the argument.
- Although we respect the opinion of the reviewer #2 that our paper doesn’t match with the objective of JCM, we do not agree with him/her. We are, in fact, convinced that, since frailty plays a crucial role in the development and in the prognosis of so many diseases, it merits a special attention by clinicians and deserves a central position in Clinical Medicine and in Clinical Medicine-related scientific Journals. In any case, we let the Editor to judge whether our review would match the objective of JCM and, of course, we will accept the final decision.
Reviewer 3 Report
Comments and Suggestions for Authors
I have to congratulate you for the work done, however, I need to point out a few questions:
Line 29: Although I understand the concept of the work, the Abstract does not make clear the possible conclusions that may have been reached through the analysis of the results after the search used. Please solve the problem.
Line 47: Although the writing is excellent and everything is very clear, putting all the information in a single paragraph in the introduction makes it a little difficult to follow the reading. I recommend separating the text into different paragraphs.
Line 132: At this point they once again focus on the definition of fragility, which has been addressed at the beginning of the introduction. Is this the correct title for this section? Wouldn't it be better to find another one that explains better what is meant? Furthermore, I encourage you not to use such long paragraphs, as I mentioned in the previous point. Which is repeated throughout the entire text.
Line 469: Why is there red text in this section? Please correct it.
Line 1678: Point 5 and point X talk about risk factors for frailty and biomarkers of frailty. Both points could be in the same section or at least next to each other. Having them so far apart makes reading sometimes difficult to follow. Furthermore, point 10 addresses how you will value... seems to repeat the concept of point 5.
Line 1876: If there is only one subsection within a section, perhaps it would be better not to include it. It is recommended if there are at least two points.
Generic comments: The work is excellent, however, due to its volume, I miss it being somewhat more organized. Sometimes it is difficult to follow and know why he is talking at each point. Perhaps in a generic way ordering the different points of the work would help in its reading. On the other hand, the non-use of different paragraphs in each of the sections makes reading difficult. I encourage you to change mainly these two points, since if this were the case, I think that the work would gain a lot.
Author Response
- First of all, we would like to thank the reviewer for his/her appreciation of our work.
- According to the reviewer’s suggestion, we have modified the Abstract by including the possible conclusions that we have reached through the analysis of the results after the search used.
- Thank you for your suggestion. We have separated the text into different paragraphs
- We accept your suggestion and we have changed the text accordingly and included additional paragraphs
- We have changed the red color of the text used in line 469.
- We have changed the text to avoid repeating the concept expressed in point 5 and 10
- We have removed the subsection od point 11.
- We try to follow the suggestion of reviewer #3 and we have ordered in a different way and re-organize the different points of the review
Round 2
Reviewer 2 Report
Comments and Suggestions for Authors
We thank the reviewer for the suggestion. We have considered the possibility to perform a meta-analysis but we believe that such approach would have not been representative of the complex nature of frailty. There are so many aspects of frailty and so many different ways to measure it, as well as so many areas and domains where the term frailty has been used to define specific conditions of vulnerability that we think almost impossible to constrain and combines all the results of the numerous studies published on this argument, into a single statistical analysis, based on strict protocols such as the PRISMA protocol. We believe that the traditional narrative review, based on the critical overview of previously research published in the literature, would represent better the complexity of the argument. Meta-analysis is not obligatory, but methodology of systematic review based on PRISMA statement improve de credibility and evidence of your research.
Although we respect the opinion of the reviewer #2 that our paper doesn’t match with the objective of JCM, we do not agree with him/her. We are, in fact, convinced that, since frailty plays a crucial role in the development and in the prognosis of so many diseases, it merits a special attention by clinicians and deserves a central position in Clinical Medicine and in Clinical Medicine-related scientific Journals. In any case, we let the Editor to judge whether our review would match the objective of JCM and, of course, we will accept the final decision.Frailty matches with the journal scope, but the type of research follows a subjective methodology, turn to systematic review improve objectivity and match with the journal style
Author Response
- We have already replied to Reviewer #2. Frailty probably deserves one systematic review for each area and domain in which is involved. We tried to do our best to examine the multiple and different aspects of frailty and to combine them in a single review to facilitate an omni comprehensive view of frailty. Taking advantage of the different expertise of all the Authors, we intended to promote a multidisciplinary holistic approach to this complex syndrome. In our opinion, such approach is required for planning a multidomain and personalized intervention strategy, aimed to slow down frailty progression and even to completely reverse the course of this condition. In addition, it is also necessary for preventing occurrence of frailty and for guaranteeing a successful ageing or ageing without disability or loss of independence.
- We would like to response to reviewer #2 that our narrative review, although not systematic, is based on the extensive analysis of the literature, with a total of 530 papers cited in the text and listed in the reference section. Based on the large and accurate literature search performed, it is our opinion that this review cannot be really considered as “limited by a subjective methodology” in the literature search.